# Copy Suppression: Comprehensively Understanding a Motif in Language Model Attention Heads

## Abstract

We present the copy suppression motif: an algorithm implemented by attention heads in large language models that reduces loss. If i) language model components in earlier layers predict a certain token, ii) this token appears earlier in the context and iii) later attention heads in the model suppress prediction of the token, then this is copy suppression. To show the importance of copy suppression, we focus on reverse-engineering attention head 10.7 (L10H7) in GPT-2 Small. This head suppresses naive copying behavior which improves overall model calibration, which explains why multiple prior works studying certain narrow tasks found negative heads that systematically favored the wrong answer. We uncover the mechanism that the negative heads use for copy suppression with weights-based evidence and are able to explain 76.9% of the impact of L10H7 in GPT-2 Small, by this motif alone. To the best of our knowledge, this is the most comprehensive description of the complete role of a component in a language model to date. One major effect of copy suppression is its role in self-repair. Self-repair refers to how ablating crucial model components results in downstream neural network parts compensating for this ablation. Copy suppression leads to self-repair: if an initial overconfident copier is ablated, then there is nothing to suppress. We show that self-repair is implemented by several mechanisms, one of which is copy suppression, which explains 39% of the behavior in a narrow task. Interactive visualizations of the copy suppression phenomena may be seen at our web app https://copy-suppression.streamlit.app/.

## 1 Introduction

Mechanistic interpretability research aims to reverse engineer neural networks into the algorithms that network components implement (Olah, 2022). A central focus of this research effort is the search for explanations for the behavior of model components, such as circuits (Cammarata et al., 2020; Elhage et al., 2021), neurons (Radford et al., 2017; Bau et al., 2017; Gurnee et al., 2023) and attention heads (Voita et al., 2019; Olsson et al., 2022). However, difficulties in understanding machine learning models has often limited the breadth of these explanations or the complexity of the components involved (Räuker et al., 2023).

In this work we explain how "Negative Heads" (which include 'negative name mover heads' from Wang et al. (2023) and 'anti-induction heads' from Olsson et al. (2022)) function on the natural language training distribution in GPT-2 Small. Previous work found that Negative Heads systematically write against the correct completion on narrow datasets, and we explain these observations as instances of **copy suppression**. Copy suppression accounts for a majority of the head's behavior and reduces the model's overall loss. To the best of our knowledge, our explanation is the most comprehensive account of the function of a component in a large language model (Section 5 reviews related literature).

We define **Negative Heads** as attention heads which primarily reduce the model's confidence in particular token completions. We show that the main role of Negative Heads in GPT-2 Small is **copy suppression** (Figure 1), which is defined by three steps:

1. **Prior copying**. Language model components in early layers directly predict that the next token is one that already appears in context, e.g that the prefix "All's fair in love and" is completed with " love".
2. **Attention**. Copy suppression heads detect the prediction of a copied token and attend back to the previous instance of this token (" love").
3. **Suppression**. Copy suppression heads write directly to the model's output to decrease the logits on the copied token.

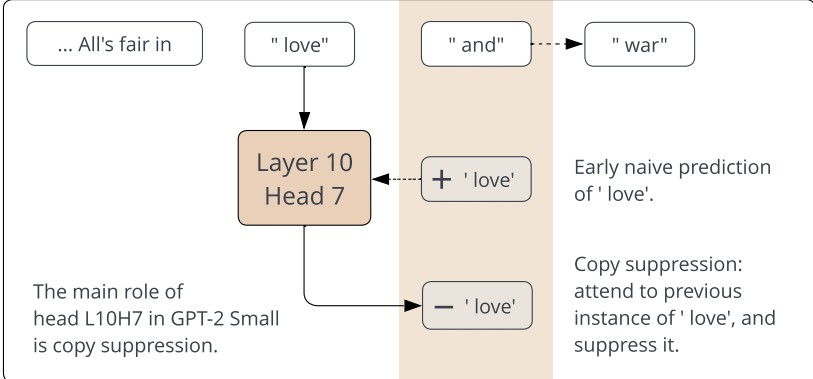

Figure 1: L10H7's copy suppression mechanism. Attention head L10H7 detects the naive prediction of "love" (copied from earlier in the prompt by upstream model components), attends back to the previous instance of the "love" token, and writes to the residual stream in the opposite direction to the "love" unembedding, thereby suppressing the prediction of that token.

By lowering incorrect logits, steps 1–3 can increase the probability on correct completions (e.g " war") and decrease model loss.[1] **Our central claim is that at least 76.9% of the role of attention head L10H7 on GPT-2 Small's training distribution is copy suppression**. However, we do not explain precisely when or how much copy suppression is activated in different contexts. Nevertheless, to the best of our knowledge, there is no prior work which has explained the main role of any component in a large language model in terms of its input stimulus and specific downstream effect across a whole training distribution.

Explaining language models components across wide distributions in mechanistic detail may be important for engineering safe AI systems. While interpreting parts of language models on narrow distributions (Hanna et al., 2023; Heimersheim and Janiak, 2023; Wang et al., 2023) may be easier than finding complete explanations, researchers can be misled by hypotheses about model components that do not generalize (Bolukbasi et al., 2021). Mechanistically understanding models could fix problems that arise from opaque training processes, as mechanisms can predict behavior on off-distribution and adversarial inputs rather than merely those that arise in training (Mu and Andreas, 2020; Goh et al., 2021; Carter et al., 2019).

Mechanistic interpretability research is difficult to automate and scale (Räuker et al., 2023), and understanding negative and backup heads[2] could

be crucial for further progress. Many approaches to automating interpretability use **ablations** - removing a neural network component and measuring the effect of this intervention (Conmy et al., 2023; Wu et al., 2023; Bills et al., 2023; Chan et al., 2022). Ideally, ablations would provide accurate measures of the importance of model components on given tasks, but negative and backup components complicate this assumption. Firstly, negative components may be ignored by attribution methods that only find the positive components that complete tasks. This means that these attribution methods will not find faithful explanations (Jacovi and Goldberg, 2020) of model behavior. Secondly, backup components may counteract the effects of ablations (Li et al., 2023; Turner et al., 2023) and hence cause unreliable importance measurements.

In this work we rigorously reverse-engineer attention head L10H7 in GPT-2 Small to show that its main role on the training distribution is copy suppression. We do not know *why* language models form copy suppression components, but in Section 4.1 and Appendix C we discuss ongoing research into some hypotheses. Appendix A provides evidence that copy suppression occurs in models trained without dropout. Our main contributions are:

1. Finding the main role of an attention head in an LLM on an entire training distribution (Section 2), and verifying this hypothesis (Section 3.3).
2. Using novel weights-based arguments to explain the role of language model components

---

[1]We recommend using our web app `https://copy-suppression.streamlit.app/` to understand L10H7's behavior interactively.

[2]We define backup heads (see Section 4) as attention heads

that respond to the ablation of a head by imitating that original behavior.

(Section 3).

3. Applying our mechanistic understanding to the practically important self-repair phenomenon, finding that copy suppression explains 39% of self-repair in one setting (Section 4).

## 2 Negative Heads Copy Suppress

In this section we show that Negative Head L10H7 suppresses copying across GPT-2 Small's training distribution. We show that copy suppression explains most of L10H7's role in the model, and defer evaluation of our mechanistic understanding to Section 3.3. We use the **logit lens** (nostalgebraist, 2020) technique to measure what intermediate model components predict, and use **mean ablation** to delete internal model activations.

### 2.1 Behavioral Results

We can find where L10H7 has the largest impact by looking at the OpenWebText (Gokaslan et al., 2019) examples where mean ablating L10H7's effect on model outputs increases loss. Specifically, we sampled from the top 5% of completions where L10H7 had greatest effect as these accounted for half of the attention head's loss reducing effect across the dataset. **80% of the sampled completions were examples of copy suppression** when we operationalized the three qualitative copy suppression steps from Section 1 by three corresponding conditions:

1. The model's predictions at the input to L10H7 included a token which appeared in context as one of the top 10 most confident completions (as measured by the logit lens, a technique to measure the direct influence of specific model components on output logits using the unembedding matrix);
2. The source token was one of the top 2 tokens in context that L10H7 attended to most;
3. The 10 tokens that L10H7 decreased logits for the most included the source token.

Examples can be found in the Section 2. These results and more can also be explored on our interactive web app (`https://copy-suppression.streamlit.app/`).

### 2.2 How Does L10H7 Affect the Loss?

To investigate the relative importance of the direct and indirect effect of L10H7 on the model's loss, we decompose its effect into a set of different paths (Elhage et al., 2021; Goldowsky-Dill et al., 2023), and measure the effect of ablating certain paths.

We measure the effect on model's loss as well as the KL divergence to the model's clean predictions. Results can be seen in Figure 2.

Fortunately, we find that most of L10H7's effect on loss was via the direct path to the final logits. This suggests that a) explaining the direct path from L10H7 to outputs would explain the main role of the attention head in the model and b) KL divergence is correlated with the increase in loss of ablated outputs. Our goal is to show that our copy suppression mechanism faithfully reflects L10H7's behaviour (Section 3.3) and therefore in the rest of our main text, we focus on minimizing KL divergence, which we discuss further in Section 3.3.1.

## 3 How Negative Heads Copy Suppress

In this section, we show that copy suppression explains 76.9% of L10H7's behavior on OpenWebText. To reach this conclusion, we perform the following set of experiments:

1. In Section 3.2, we analyse the output-value (OV) circuit, which is the circuit determining what information the attention head moves from source to destination tokens. We show that the head suppresses the prediction of 84.70% of tokens which it attends to.
2. In Section 3.2, we analyse the query-key (QK) circuit, which is the circuit determining which tokens the head will pay attention to. We show that the head attends to the token which the model is currently predicting across 95.72
3. In Section 3.3, we define a form of ablation (CSPA) which deletes all of L10H7's functionality except 1. and 2., and preserves 76.9% of its effect.

In step 3 we project L10H7's outputs onto the unembedding vectors, but apply a filtering operation (that is weaker than a weights-based projection) to the QK circuit, as described in Section 3.3.1. We also performed an ablation that involved projecting the query vectors onto unembedding vectors present in the residual stream (Appendix M), but found that this did not recover as much KL divergence, likely due to issues discussed in Section 4. In Section 3.1-3.2 we apply the zeroth MLP layer of GPT-2 Small to its embedding, ie we use $\mathrm{MLP}_0(W_E)$ rather than $W_E$ and call this the model's '**effective embedding**'. We discuss this in Appendix H and compare with other works.

| Prompt | Source token | Incorrect completion | Correct completion |
|---|---|---|---|
| ... Millions of **Adobe** users picked easy-to-guess ~~**Adobe**~~ **passwords** ... | " Adobe" | " Adobe" | " passwords" |
| ... tourist area in **Beijing**. A university in ~~**Beijing**~~ **Northeastern** ... | " Beijing" | " Beijing" | " Northeastern" |
| ... successfully stopped **cocaine** and ~~**cocaine**~~ **alcohol** ... | " cocaine" | " cocaine" | " alcohol" |

Table 1: Dataset examples of copy suppression, in cases where copy suppression behaviour decreases loss by suppressing an incorrect completion.

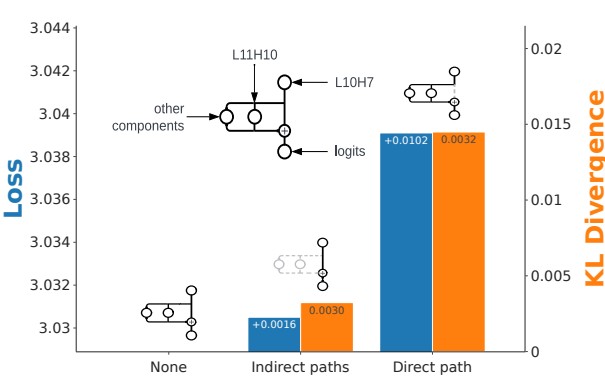

Figure 2: Loss effect of L10H7 via different paths. Grey paths denote ablated paths.

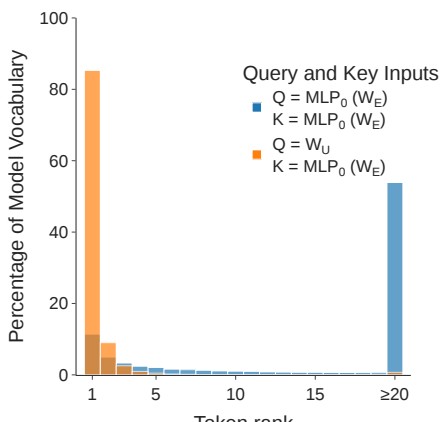

Figure 3: Distribution of ranks of diagonal elements of Eqn. (2).

### 3.1 OV Circuit

To understand L10H7's output, we study the simple setting where the attention head i) only attends to a single source token and ii) the source token position only contains information about one token. We can then look at what effect L10H7 has on the model's logits for each token in the vocabulary. This motivates studying L10H7's OV circuit (Elhage et al., 2021), with our effective embedding refinement: $W_U W_{OV}^{L10H7} \text{MLP}_0(W_E) \in \mathbb{R}^{n_{\text{vocab}} \times n_{\text{vocab}}}$ (1), where $W_U$ and $\text{MLP}_0(W_E)$ is the unembedding and effective embedding matrix of the model, respectively, and $W_{OV}^{L10H7}$ is the OV Matrix of L10H7.

The OV circuit (1) studies the impact that L10H7 has on all output tokens, given it attended to the effective embedding of a particular input token. The $i$th column of (1) is the vector of logits added at any destination token which attends to the $i$th token in the model's vocabulary (ignoring layernorm scaling). If L10H7 is suppressing the tokens that it attends to, then the diagonal elements of (1) will consistently be the most negative elements in their columns. This is what we find: 84.70% of the

tokens in GPT-2 Small's vocabulary have their diagonal elements as one of the top 10 most negative values in their columns, and 98.86% of tokens had diagonal elements in the bottom 5%. This suggests that L10H7 is copy suppressing almost all of the tokens in the model's vocabulary.

This effect can also be seen in practice. We filtered for (source, destination token) pairs in OpenWebText where attention in L10H7 was large, and found that in 78.24% of these cases the source was among the 10 most suppressed tokens from the direct effect of L10H7 (full experimental details in Appendix E). This indicates that our weights-based analysis of L10H7's OV circuit does actually tell us about how the head behaves on real prompts.

### 3.2 QK Circuit

Having understood L10H7's outputs in a controlled setting, we need to understand when the head is activated by studying its attention patterns. In a similar manner to Section 3.1 we study L10H7's attention in the simple setting where i) the query input is equal to the unembedding vector for a single token and ii) the key input is the effective embedding for another single token, i.e we study the QK cir-

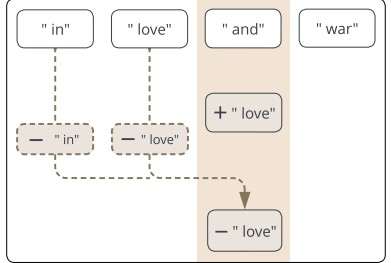

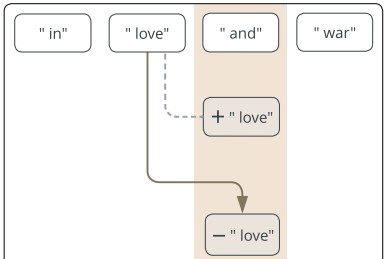

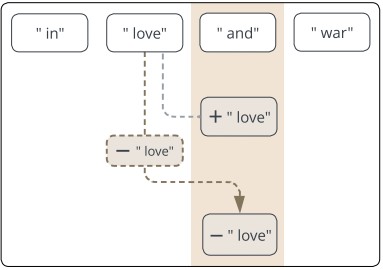

## OV Ablation

Project each result vector along the unembedding vector for that token (and take only the negative components).

## QK Ablation

Mean ablate all vectors, except from source tokens which are most strongly predicted at the destination token.

## Copy Suppression Preserving Ablation (CSPA)

Both OV and QK ablations.

Figure 4: Illustration of three different kinds of ablation: just OV, just QK, and CSPA.

cuit $W_U W_{QK}^{\text{L10H7}} \text{MLP}_0(W_E) \in \mathbb{R}^{n_{\text{vocab}} \times n_{\text{vocab}}}$ (Eqn. (2)).[3]

Copy suppression (Section 1) suggests that L10H7 has large attention when i) a token is confidently predicted at the query position and ii) that token appeared in the context so is one of the key vectors. Therefore we expect the largest elements of each row of Eqn. (2) to be the diagonal elements of this matrix. Indeed, in Figure 3 (orange bars) we find that 95.72% of diagonal values in this matrix were the largest in their respective rows.

However, this result alone doesn't imply that copying (the first step of the three copy suppression steps in Section 1) explains L10H7's attention. This is because GPT-2 Small uses the same matrix for embeddings and unembeddings, so L10H7 could simply be matching similar vectors at query and keyside (for example, in a 'same matching' QK matrix (Elhage et al., 2021)) Therefore in Figure 3 (blue bars) we also compare to a baseline where both query and keys are effective embeddings,[4] and find that the ranks of the diagonal elements in their rows are much smaller, which provides evidence that $W_{QK}^{\text{L10H7}}$ is not merely a 'same matching' matrix. We also verify the copy suppression attention pattern further in Appendix L.1. However, one limitation of our analysis of the QK circuit is that this idealised setup does not completely faithfully represent L10H7's real functioning (Appendices L.2, L.3 and M).

### 3.3 How much of L10H7's behavior have we explained?

In this section, we perform an ablation which deletes all functionality of L10H7's OV and QK circuits, except for the mechanisms described in Section 3.1 and 3.2 respectively, with the goal of seeing how much functionality we can remove *without* damaging performance. We refer to this as **Copy Suppression-Preserving Ablation** (CSPA). In the Section 3.3.1 section we explain exactly how each part of CSPA works, and in the Section 3.3.2 section we present the ablation results.

#### 3.3.1 Methodology

CSPA consists of both an **OV ablation** and a **QK ablation**.

**OV ablation**. The output of an attention head at a given destination token $D$ can be written as a sum of result vectors from each source token $S$, weighted by the attention probabilities from $D$ to $S$ (Elhage et al., 2021). We can project each of these vectors onto the unembedding vector for the corresponding source token $S$. We only keep negative components.[5]

**QK ablation**. We mean ablate the result vectors from each source token $S$, except for the top 5% of source tokens which are predicted with highest probability at the destination token $D$ (as measured with the logit lens).

As an example of how the OV and QK ablations work in practice, consider the opening example "All's fair in love and war". In this case the destination token $D$ is " and". The token "love" is highly predicted to follow $D$ (as measured with the logit lens), and also appears as a source token $S$, and so we would take the result vector from $S$ and project it onto the unembedding vector for

---

[3]We ignore bias terms in the key and query parts (as we find that they do not change results much in Appendix L). Our experimental setup allows us to ignore LayerNorm (Appendix G).

[4]i.e in Eqn. (2) we replace the $W_U$ term with $\text{MLP}_0(W_E)$.

[5]In Figure 16 we show the results when we also keep positive components.

" love", mean-ablating everything else. Although this deletes most of the dimensions of L10H7, it still captures how L10H7 suppresses the " love" prediction.

**Ablation metric**. After performing an ablation, we can measure the amount of L10H7's behavior that we have explained by comparing the ablation to a baseline that mean ablates L10H7's direct effect. Formally, if the model's output token distribution on a prompt is $\pi$ and the distribution under an ablation Abl is $\pi_{\text{Abl}}$, then we measure the KL divergence $D_{\text{KL}}(\pi||\pi_{\text{Abl}})$. We average these values over OpenWebText for both ablations we use, defining $\overline{D_{\text{CSPA}}}$ for CSPA and $\overline{D_{\text{MA}}}$ for the mean ablation baseline. Finally, we define the effect explained as $1 - \left( \overline{D_{\text{CSPA}}} / \overline{D_{\text{MA}}} \right)$ (Eqn. (3)).

We choose KL divergence for several reasons, including how 0 has a natural interpretation as the ablated and clean distributions being identical – in other words, 100% of the head's effect being explained by the part we preserve. See Appendix I for limitations, comparison and baselines.

### 3.3.2 Results

CSPA explains 76.9% of L10H7's behavior. Since the QK and OV ablations are modular, we can apply either of them independently and measure the effect recovered. We find that performing only the OV ablation leads to 81.1% effect explained, and only using QK leads to 95.2% effect explained. To visualize the performance of CSPA, we group

each OpenWebText completion into one of 100 percentiles, ordered by the effect that mean ablation of L10H7 has on the output's KL divergence from the model. The results are shown in Figure 6, where we find that CSPA preserves a larger percentage of KL divergence in the cases where mean ablation is most destructive: in the maximal percentile, CSPA explained 88.1% of L10H7's effect.

## 4 Applications of Copy Suppression

In this section, we explore some different applications of copy suppression. First, we connect it to the previously observed phenomena of anti-induction, while also providing evidence that it occurs in several different sizes and classes of models. Second, we discuss the phenomena of self-repair, which refers to how neural network components can sometimes compensate for perturbations made to earlier components.

We will focus on the narrow Indirect Object Identification (IOI; Wang et al. (2023)) task during this section. We give a short introduction to IOI in points i)-iii) below. Non-essential further details can be found in Wang et al. (2023).

- i) The IOI task consists of sentences such as 'When John and Mary went to the store, Mary gave a bottle of milk to' which are completed with the indirect object (IO) ' John'.
- ii) The task is performed by an end-to-end circuit. The final attention heads involved in this circuit are called Name Mover Heads; they copy the IO to the model's output.
- iii) We can measure the extent to which IOI occurs by measuring the logit difference metric, which is equal to the difference between the ' John' and ' Mary' logits in the above example.

Copy suppression heads like L10H7 usually come after the name mover heads. They detect the IO prediction, attend back to the first instance of the IO, and suppress it (but not enough to change the model's prediction). This is a relatively clean domain in which to study copy suppression.

### 4.1 Anti-induction

While studying induction heads, Olsson et al. (2022) discovered attention heads which identify repeating prefixes and suppress the prediction of the token which followed the first instance of the prefix - in other words the opposite of the induction pattern. We suspected this anti-induction was an instance of copy suppression, because induction heads writing the prediction of this token into the residual stream could cause copy suppression heads to attend back to and suppress the first instance of the token. To investigate this, we created *scores* for how much a set of attention heads (across GPT, Pythia and SoLU architectures copy suppressed on both the IOI task and the anti-induction task. We measured these *scores* by taking the negation of the attention head's direct effect on the correct token: in the induction task this was the repeated token, in the copy-suppression task this was the indirect object name. We found a strong correlation in the quadrant where both were positive (Figure 5).

There are two important lessons to draw from these experiments. Firstly, **copy suppression heads exist in larger models, and models of different classes**. We observed copy suppression heads in models as large as Pythia-6B. Secondly, this result demonstrates the danger of drawing conclusions from narrow distribution-based studies, since it strongly implies that two seemingly sep-

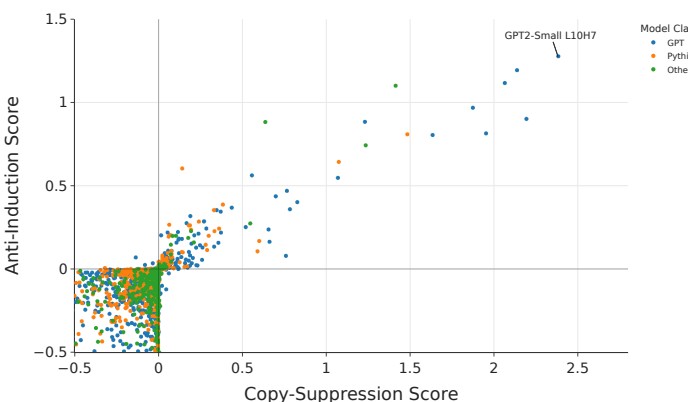

Figure 5: Anti-induction and copy suppression on the IOI task compared.

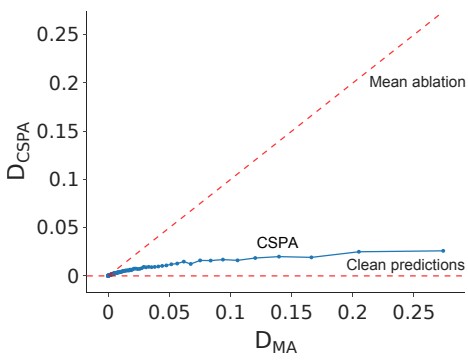

Figure 6: We plot $(\overline{D_{\text{CSPA}}}, \overline{D_{\text{MA}}})$ for each percentile of our OpenWebText data (with percentiles given by the values of $D_{\text{MA}}$).

| Head Type | Response to Name Movers predicting $T$ | Effect of attending to $T$ |
|-----------|------------------------------------------|-----------------------------|
| Negative | **More** attention to $T$ | **Decrease** logits on $T$ |
| Backup | **Less** attention to $T$ | **Increase** logits on $T$ |

Table 2: Qualitative differences between Negative and Backup Heads.

arate and task-specific behaviors (anti-induction on random repeated sequences, and suppression of the IO token in the IOI task) are actually not task-specific at all, but are both consequences of the same core algorithm: copy suppression. Studying attention heads on just one of these distributions might give the incorrect impression that it was using details of the task to make its predictions, but our study across the entire OWT distribution has revealed an algorithm which explains both behaviours.

### 4.2 Self-Repair

Self-repair refers to how some neural network components compensate for other components that have been perturbed earlier in the forward pass (Mc-Grath et al., 2023). Copy suppressing components self-repair: if perturbing specific model components causes them to stop outputting an unembedding, copy suppression is deactivated. In this section, we show that copy suppression explains 39% of self-repair in one setting. However Appendix R gives weights-based evidence that self-repair relies on more than just copy suppression, and finds that the unembedding direction in the residual stream does not have a large effect on self-repair.

To visualize self-repair under an ablation of the three Name Mover Heads, for every attention head downstream of the Name Mover Heads we measure its original contribution to logit difference ($x_c$),

as well as its contribution to logit difference post-ablation ($y_c$). We then plot all these ($x_c, y_c$) pairs in Figure 8.

In Figure 8, the higher the points are above the $y = x$ line, the more they contribute to self-repair. This motivates a way to measure self-repair: if we let $C$ denote the set of components downstream of Name Mover Heads and take $c \in C$, then the proportion of self-repair that a component $c$ explains is $(y_c - x_c) / \sum_{i \in C}(y_i - x_i)$ (Eqn. (4)). The sum of the proportions of self-repair explained by Negative Heads L10H7 and L11H10 is 39%. This proportion is almost entirely copy suppression since Appendix O shows that the Negative Heads in the IOI task are entirely modulated by Name Mover Heads.

However, Figure 8 indicates another form of self-repair in the heads on the right side of the figure: these heads do not have large negative effects in clean forward passes, but then begin contributing to the logit difference post-ablation. We found that these backup heads on the right hand side use a qualitatively different mechanism for self-repair than (copy suppressing) negative heads, which we summarise behaviorally in Table 2.

To justify the description in Table 2, we analyze how Name Movers determine the attention patterns of self-repairing heads using Q-composition, i.e. their queries are computed from the output of upstream attention heads. We study Q-composition

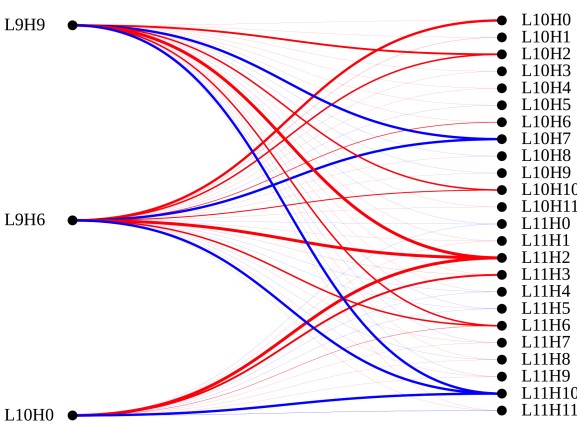

Figure 7: Red edges denote less, and blue edges denote more attention to names due to the Name Movers.

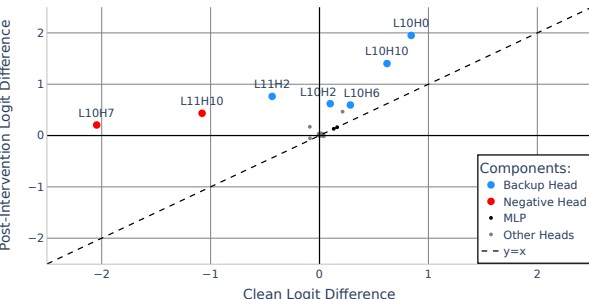

Figure 8: Ablating the Name Mover Heads in Layer 9 causes a change in the direct effects of all the downstream heads. Plotting the Clean Logit Difference vs the Post-Intervention Logit Difference for each head highlights the heads above the $y = x$ line which perform self-repair.

between a Name Mover's OV matrix $W_{OV}$ and the QK matrix $W_{QK}$ of downstream heads by calculating $\mathrm{MLP}_0(W_E)^\top W_{OV}^\top W_{QK}\mathrm{MLP}_0(W_E)$ and find that backup heads attend *less* to names when Name Movers copy them, and negative heads attend more (Figure 7; Appendix N). Combining this result with the prior results that i) backup heads copy names (Wang et al., 2023) and ii) negative heads have negative-copying OV matrices (Section 3.1), this explains self-repair at a high-level in IOI: when the Backup/Negative heads attend more/less to a token $T$ upon the Name Mover's ablation, they *copy more*/*suppress less* of $T$, increasing the logit difference and thus self-repairing. However, there are limits to this line of reasoning, since in Appendix R we explore how the unembedding component does not seem to be the most important component used; we hope future works can probe self-repair further.

## 5 Related Work

**Explanations of neural network components** in post-hoc language model interpretability include explanations of neurons, attention heads and circuits. Related work includes the automated approach by Bills et al. (2023) and manual explanations found by Voita et al. (2023) who both find suppression neurons. More comprehensive explanations are found in Gurnee et al. (2023). Attention heads correlated with previous tokens (Vig, 2019) and rare words (Voita et al., 2019) have been analyzed. Circuits have been found on narrow distributions (Wang et al., 2023) and induction heads (Elhage et al., 2021) are the most general circuits found in language models, but they have only been explained in as much detail as our work in toy models. Chan et al. (2022)'s loss recovered metric inspired our loss recovered analysis.

**Iterative inference**. Greff et al. (2017) propose that neural networks layers iteratively update feature representations rather than recomputing them, in an analysis specific to LSTMs and Highway Networks. Several works have found that transformer language model predictions are iteratively refined (Dar et al., 2022; nostalgebraist, 2020; Belrose et al., 2023; Halawi et al., 2023) in the sense that the state after intermediate layers forms a partial approximation to the final output, though no connections have yet been made to Negative Heads.

## 6 Conclusion

In summary, in this work we firstly introduced **copy suppression**, a description of the main role of an attention head across GPT-2 Small's training distribution. Secondly, we applied weights-based arguments using QK and OV circuits to mechanistically verify our hypotheses about copy suppression. Finally, we showed how our comprehensive analysis has applications to open problems in ablation-based interpretability (Section 4).

Two limitations of our work include our understanding of the query inputs to self-repair heads, and the transferability of our results to different models. In both Section 3.2 and 4 we found that copy suppression and self-repair rely on more than simply unembedding directions, and we hope that future work can fully explain this observation. Further, while we show that some of our insights generalize to large models (Section 4.1 and A), we don't have a mechanistic understanding of copy suppression in these cases. Despite this, our work shows that it is possible to explain LLM components across broad distributions with a high level of detail. For this reason, we think that our insights will be useful for future interpretability research.

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

## Glossary 

**Anti-induction** Anti-induction heads are our name for 'anti-copying prefix search' heads (Olsson et al., 2022). See Section 4.1.

**Backup heads** are attention heads that are characterised by responding to the ablation of a head by imitating the original behavior, studied in the IOI task in Section 4.

**Copy Suppression** is a mechanism in a language models determined by the three steps **naive copying**, **attention** and **suppression**, as described in Section 1.

**Copy suppression-preserving ablation (CSPA)** refers to our ablation that deletes all functionality of attention head 10.7 except the copy suppression mechanism (Section 3.3.1).

**Direct Logit Attribution** is defined in https://www.neelnanda.io/mechanistic-interpretability/glossary.

**Effective embedding** is what models use to identify tokens at different positions after the first transformer layer. We define this as $\text{MLP}_0(W_E)$, and discuss the choice in Appendix H.

**Eqn. (1)** is defined in Section 3.1 and is our OV circuit expression.

**Eqn. (2)** is defined in Section 3.2 and is our QK circuit expression.

**Eqn. (3)** is defined in Section 3.3.1 and measures how well ablations preserve L10H7's functionality.

**Eqn. (4)** is defined in Section 4.2 and measures how much self-repair a component $c$ explains.

**Induction heads** are attention heads that identify repeating prefixes, attend back to the token following the previous instance of the prefix, and predict that same token will come next in the sequence.

**IOI** . The IOI task is the prediction that ' John' completes the sentence 'When John and Mary went to the store, Mary gave a bottle of milk to' (Wang et al., 2023).

**Logit difference** is described in point iii) in Section 4.2.

**Logit Lens** We can measure which output predictions different internal components push for by applying the Logit Lens method (nostalgebraist, 2020). Given model activations, such as the state of the residual stream or the output of an attention head, we can multiply these activations by GPT-2 Small's unembedding matrix. This measures the direct effect (ie not mediated by any downstream layers) that this model component has on the output logits for each possible token in the model's vocabulary (sometimes called direct logit attribution). The Logit Lens method allows us to refer to the model's predictions at a given point in the network.

**Mean ablation** refers to replacing the output of a machine learning model component with the mean output of that component over some distribution.

**Name Mover Heads** are heads that attend to (and copy) IO rather than S in the IOI task.

**Negative Head** are attention heads in transformer language models which which primarily reduce the model's confidence in particular token completions. This is a qualitative definition. These heads tend to be rare since the majority of attention heads in models positively copy tokens (Elhage et al., 2021; Olsson et al., 2022).

**Self-repair** refers to how some neural network components compensate for other components that have been perturbed earlier in the forward pass (McGrath et al., 2023).

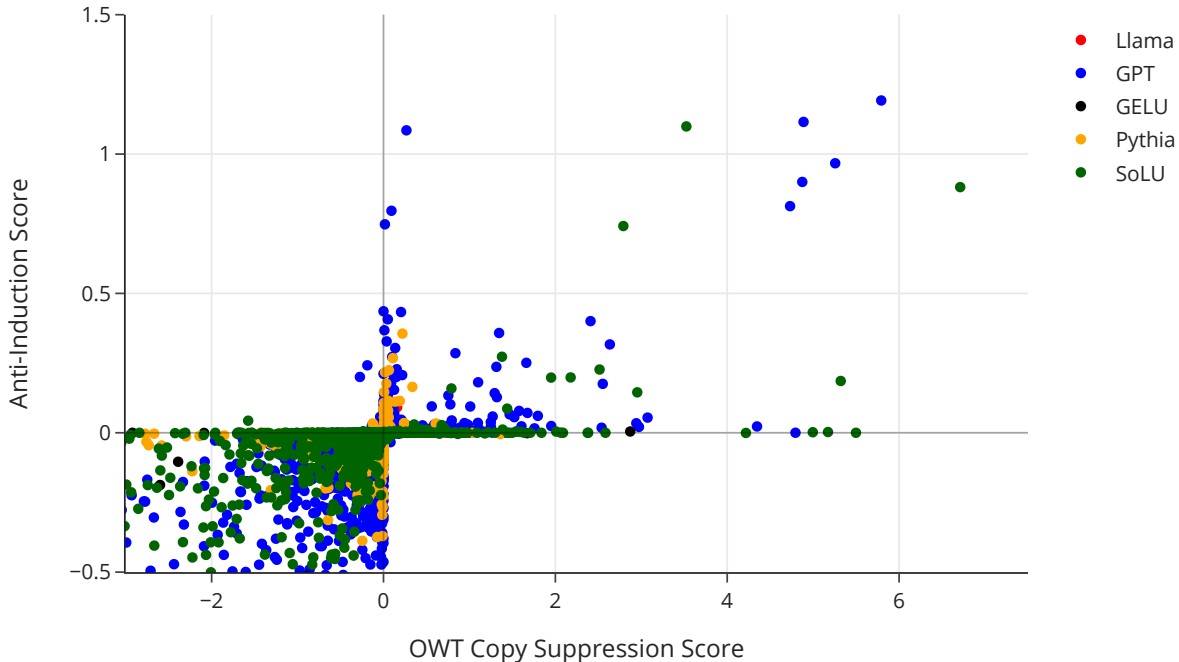

Figure 9: Copy Suppression scores on OWT measured against the Anti-Induction scores in the IOI distribution.

## A    Scaling Copy Suppression

In this appendix we discuss how our observations about copy suppression scale to larger models (Llama-2 7B and 13B (Touvron et al., 2023)). Our high-level takeaways are that

1. General distribution copy-suppression heads exist across model scales and architectures.

2. Larger models have weaker copy suppression heads.

3. The mechanism behind the IOI task does not generalize to larger models.

**1**: Repeating the methodology that generated Figure 5, we can also compare the copy suppression effect on OWT to the anti-induction score.

We filter for token positions where there the maximally predicted token (measured via the Logit Lens) occurs in context as a token so that copy suppression is indeed a potential behavior, and again measure the direct logit attribution from the token in context.

The results are in Figure 9 and show that once more anti-induction heads do not perform any positive behavior (there are no points in bottom right or top left quadrant). We do find that the there are heads that only implement anti-induction or copy suppression, however. We discuss Llama in **2**.

**2**: In Figure 10(a) we show that while there do exist Copy Suppression heads in Llama-2 (e.g the points closest to the top right are L26H28 and L30H24 in Llama-2 7B and 13B respectively), the direct logit attribution magnitude is much smaller than in Figure 9. This suggests that the more attention heads models have, the more they distribute behavior across heads. We also find heads that copy suppress on the general distribution but not on the anti-induction task, showing further specialization.

**3**: When we studied the IOI direct logit attribution of Llama-2 7B and Llama-2 13B, we found that the direct logit attribution was smaller still, and further there was no division between positive heads and negative heads. This suggests that IOI is performed qualitatively differently to small models, perhaps not using direct attention back to the IO name.

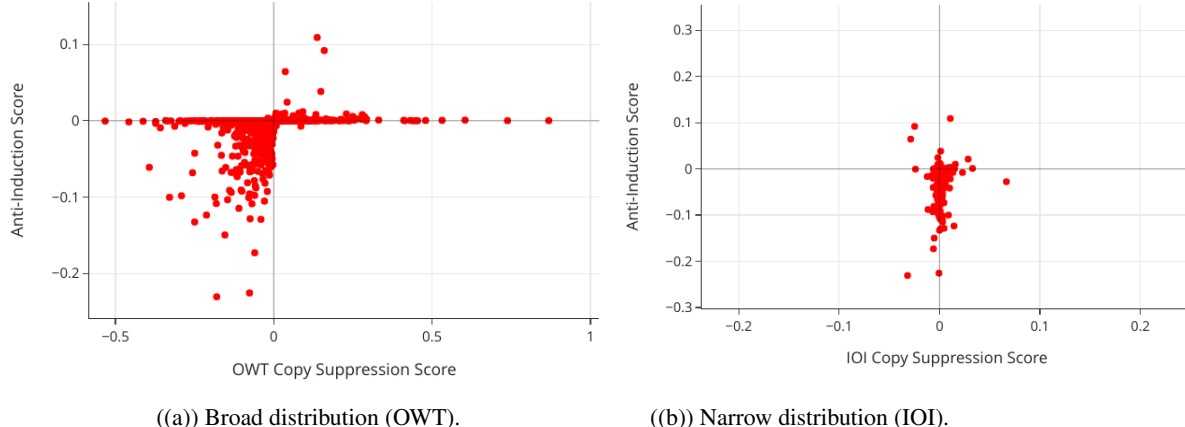

((a)) Broad distribution (OWT).      ((b)) Narrow distribution (IOI).

Figure 10: Copy Suppresion in Llama-2.

## B  L11H10

In Section 2.2 we showed that the majority of L10H7's effect on loss is via its direct effect. In this appendix we show that we can explain up to half of L10H7's indirect effect by considering the indirect through L11H10, the second Negative Head in GPT-2 Small. We repeat the same methodology as in the indirect path experiment in Figure 2, but also controlling for the path from L10H7 to L11H10 by not mean ablating this connection. We show the results in Figure 11.

The indirect path through L11H10 is special because both Negative Heads perform copy suppression, which is a self-repair mechanism: once a predicted token is suppressed, it is no longer predicted, and therefore does not activate future copy suppression components. This means that ablating head L10H7 will often result in it being backed up by head L11H10. In an experiment that ablates the effect of L10H7 on L11H10 but not on the final model output, we would expect excessive copy suppression to take place since i) L10H7 will have a direct copy suppression effect, and ii) L11H10 will copy suppress more than in normal situations, since its input from L10H7 has been ablated. Indeed the loss increase is roughly twice as large in the normal indirect effect case compared to when we control for the effect through L11H10 (Figure 11). However, surprisingly there is little effect on KL Divergence.

## C  Entropy and Calibration

A naive picture of attention heads is that they should all reduce the model's entropy (because the purpose of a transformer is to reduce entropy by concentrating probability mass in the few most likely next tokens). We can calculate a head's direct contribution to entropy by measuring (1) the entropy of the final logits, and (2) the entropy of the final logits with the head's output subtracted. In both cases, the negative head L10H7 stands out the most, and the other negative heads L11H10 and L8H10 are noticeable.

We can also examine each attention head's effect on the model's calibration. Hu et al. (2023) use **calibration curves** to visualise the model's degree of calibration. From this curve, we can define an **overconfidence metric**, calculated by subtracting the perfect calibration curve from the model's actual calibration curve, and taking the normalized $L_2$ inner product between this curve and the curve we get from a perfectly overconfident model (which only ever makes predictions of absolute certainty). The $L_2$ inner product can be viewed as a measure of similarity of functions, so this metric should tell us in some sense how overconfident our model is: the value will be 1 when the model is perfectly overconfident, and 0 when the model is perfectly calibrated. Figure 13 illustrates these concepts.

We can then measure the change in overconfidence metric from ablating the direct effect of an attention head, and reverse the sign to give us the head's direct effect on overconfidence. This is shown in the figure below, with the change shown relative to the model's original overconfidence (with no ablations). Again, we see that head L10H7 stands out, as do the other two negative heads. Interestingly, removing the direct

815
816
817
818
819
820
821
822
823
824
825
826
827
828
829
830
831
832
833
834
835
836
837
838
839
840
841
842
843
844
845
846
847

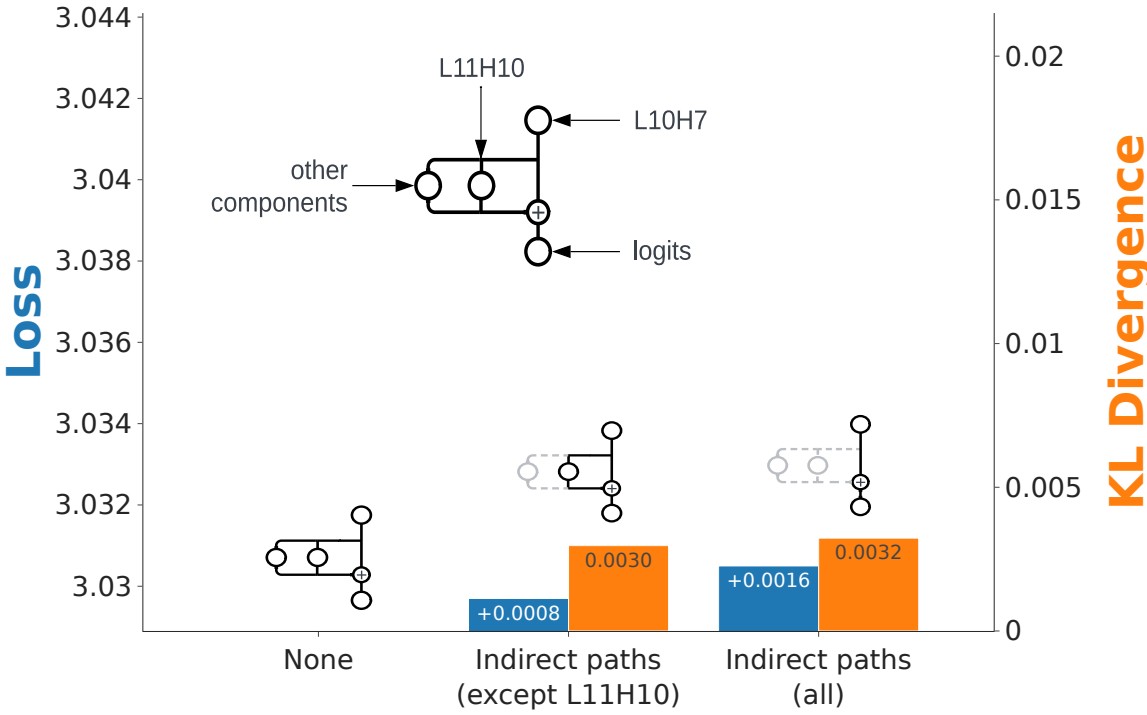

Figure 11: Loss effect of L10H7 via different paths. Grey paths denote ablated paths.

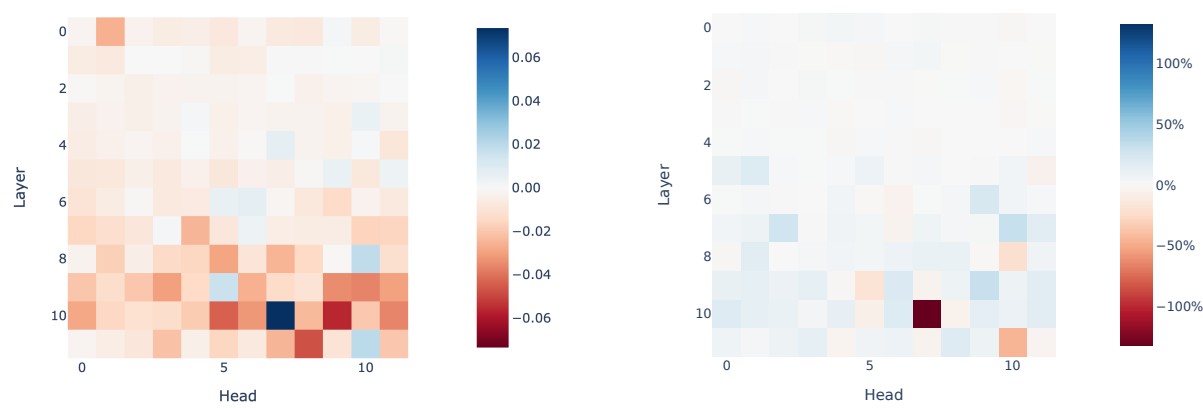

((a)) Marginal contribution to entropy (via the direct path) per head. L10H7 increases entropy (as do other negative heads like L11H10); most other heads decrease it.

((b)) Marginal effect on overconfidence metric per head. L10H7 decreases overconfidence; most other heads increase it.

Figure 12: Effect of attention heads on entropy & calibration.

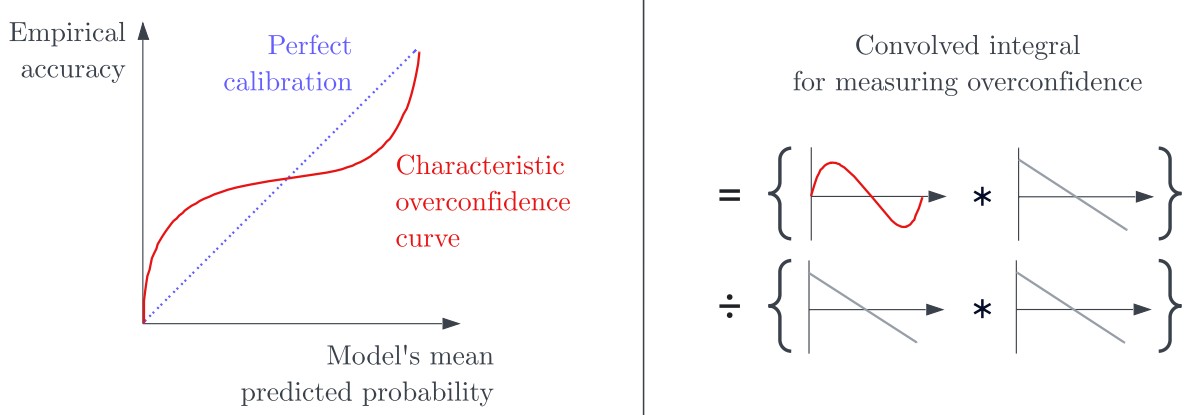

Figure 13: Illustration of the calibration curve, and overconfidence metric.

effect of head L10H7 is enough to push the model from net over-confident to net under-confident.          848

What are we to interpret from these results? It is valuable for a model to not be over-confident, because          849
the cross-entropy loss will be high for a model which makes high-confidence incorrect predictions. One          850
possible role for negative heads is that they are reducing the model's overconfidence, causing it to make          851
fewer errors of this form. However, it is also possible that this result is merely incidental, and not directly          852
related to the reason these heads form. For example, another theory is that negative heads form to          853
suppress early naive copying behaviour by the model, and in this case they would be better understood as          854
copy-suppression heads rather than "calibration heads". See the next section for more discussion of this.          855

## D   Why do negative heads form? Some speculative theories          856

This paper aimed to mechanistically explain what heads like L10H7 do, rather than to provide an          857
explanation for why they form. We hope to address this in subsequent research. Here, we present three          858
possible theories, present some evidence for/against them, and discuss how we might test them.          859

- **Reducing model overconfidence.**          860
    - **Theory**: Predicting a token with extremely high confidence has diminishing returns, because          861
      once the logprobs are close to zero, any further increase in logits won't decrease the loss if the          862
      prediction is correct, but it will increase loss if the prediction is incorrect. It seems possible that          863
      negative heads form to prevent this kind of behaivour.          864
    - **Evidence**: The results on calibration and entropy in Appendix C provide some evidence for this          865
      (although these results aren't incompatible with other theories in this table).          866
    - **Tests**: Examine the sequences for which this head decreases the loss by the most (particularly          867
      for checkpointed models, just as the negative head is forming). Are these cases where the          868
      incorrect token was being predicted with such high probability that it is in this "diminishing          869
      returns" window?          870

- **Suppressing naive copying.**          871
    - **Theory**: Most words in the English language have what we might term the "update property"          872
      - the probability of seeing them later in a prompt positively updates when they appear. Early          873
      heads might learn to naively copy these words, and negative heads could form to suppress this          874
      naive behaviour.          875
    - **Evidence**: The "All's fair in love and love" prompt is a clear example of this, and provides          876
      some evidence for this theory.          877
    - **Tests**: Look at checkpointed models, and see if negative heads form concurrently with the          878
      emergence of copying behaviour by other heads.          879

- **Suppressing next-token copying for tied embeddings.**
  - **Theory**: When the embedding and unembedding matrices are tied, the direct path $W_U W_E$ will have large diagonal elements, which results in a prediction that the current token will be copied to the next sequence position. Negative heads could suppress this effect.
  - **Evidence**: This wouldn't explain why negative heads appear in models without tied embeddings (although it might explain why the strongest negative heads we found were in GPT-2 Small, and the Stanford GPT models, which all have tied embeddings).
  - **Tests**: Look at attention patterns of the negative head early in training (for checkpointed models, with tied embeddings). See if tokens usually self-attend.

While discussing these theories, it is also important to draw a distinction between the reason a head forms during training, and the primary way this head decreases loss on the fully trained model - these two may not be the same. For instance, the head seems to also perform semantic copy suppression (see Appendix J), but it's entirely possible that this behaviour emerged after the head formed, and isn't related to the reason it formed in the first place.

## E    Experiment details for OV-Circuit in practice

We ran a forward pass on a sample of OpenWebText where we i) filtered for all (source, destination) token pairs where the attention from destination to source is above some threshold (we chose 10%), ii) measured the direct logit attribution of the information moved from each of these source tokens to the corresponding destination token and finally iii) performed the same analysis as we did in Section 3.1 - measuring the rank of the source token amongst all tokens.

We found that the results approximately matched our dynamic analysis (with slightly more noise): the proportion of (source, destination) token pairs where the source token was in the top 10 most suppressed tokens was 78.24% (which is close to the static analysis result of 84.70%).

## F    Function Words

In Section 3.1 we found that a large fraction of the tokens which failed to be suppressed were function words. The list of least copy suppressed tokens are: [' of', ' Of', ' that', ' their', ' most', ' as', ' this', ' for', ' the', ' in', ' to', ' a', 'Their', ' Its', 'When', ' The', ' its', ' these', 'The', 'Of', ' it', ' nevertheless', ' an', '<|endoftext|>', 'Its', ' have', ' some', ' By']. Sampling randomly from the 3724 tokens other than 92.59% that are copy suppressed, many are also connectives (and rarely nouns): [' plainly', ' utterly', ' enhance', ' obtaining', ' entire', ' Before', 'eering', '.)', ' holding', ' unnamed'].

It is notable that this result is compatible with all three theories which we presented in the previous section.

- **Reducing model overconfidence**. The unembedding vectors for function words tend to have smaller magnitude than the average token in GPT-2 Small. This might lead to less confident predictions for function words than for other kinds of tokens.

- **Suppressing naive copying**. There would be no reason to naively copy function words, because function words don't have this "update property" - seeing them in a prompts shouldn't positively update the probability of seeing them later. So there is no naive copying which needs to be suppressed.

- **Suppressing next-token copying for tied embeddings**. Since function words' unembedding vectors have smaller magnitudes, the diagonal elements of $W_U W_E$ are small anyway, so there is no risk of next-token copying of function words.

## G    Model and Experiment Details

All of our experiments were performed with Transformer Lens (Nanda and Bloom, 2022). We note that we enable all weight processing options,[6] which means that transformer weight matrices are rewritten

---

[6]That are described here: `https://github.com/neelnanda-io/TransformerLens/blob/main/further_comments.md#weight-processing`

so that the internal components are different and simpler (though the output probabilities are identical). For example, our Layer Norm functions only apply normalization, with no centering or rescaling (this particular detail significantly simplifies our Logit Lens experiments).

## H Effective Embedding

GPT-2 Small uses the same matrix in its embedding and unembedding layers, which may change how it learns certain tasks.[7] Prior research on GPT-2 Small has found the counter-intuitive result that at the stage of a circuit where the input token's value is needed, the output of MLP0 is often more important for token predictions than the model's embedding layer (Wang et al., 2023; Hanna et al., 2023). To account for this, we define the effective embedding. The effective embedding is purely a function of the input token, with no leakage from other tokens in the prompt, as the attention is ablated.

Why choose to extend the embedding up to MLP0 rather than another component in the model? This is because **if we run forward passes with GPT-2 Small where we delete $W_E$ from the residual stream just after MLP0 has been added to the residual stream, cross entropy loss *decreases*.**[8] Indeed, we took a sample of 3000 documents of at least 1024 tokens from OpenWebText, took the loss on their first 1024 positions, and calculated the average loss. The result was 3.047 for GPT-2 and 3.044 when we subtracted $W_E$.

## I CSPA Metric Choice

### I.1 Motivating KL Divergence

To measure the effect of an ablation, we primarily focused on the KL divergence $D_{KL}(P\|Q) = \sum_i p_i \log p_i/q_i$, where $P$ was the clean distribution and $Q$ was the distribution after our ablation had been applied. Conveniently, a KL Divergence of 0 corresponds to perfect recovery of model behavior, and it is linear in the log-probabilities $\log q_i$ obtained after CSPA.

There are flaws with the KL divergence metric. For example, if the correct token probability is very small, and a head has the effect of changing the logits for this token (but not enough to meaningfully change the probability), this will affect loss but not KL divergence. Our copy suppression preserving ablation on L10H7 will not preserve situations like these, because it filters for cases where the suppressed token already has high probability. Failing to preserve these situations won't change how much KL divergence we can explain, but it will reduce the amount of loss we explain. Indeed, the fact that the loss results appear worse than the KL divergence results is evidence that this is happening to some extent. Indeed empirically, we find that density of points with KL Divergence close to 0 but larger change in loss is greater than the opposite (change in loss close to 0 but KL larger) in Figure 14, as even using two standard deviations of change on the $x$ axis leads to more spread acrosss that axis. In Appendix I.2 we present results on loss metrics to complement our KL divergence results, and we compare these metrics to baselines in Appendix I.3.

### I.2 Comparing KL Divergence and Loss

In Figure 2, we use two different metrics to capture the effect and importance of different model components. Firstly, the amount by which ablating these components changes the average cross-entropy loss of the model on OpenWebText. Secondly, the KL Divergence of the ablated distribution to the model's ordinary distribution, again on OpenWebText. In essence, the first of these captures how useful the head is for the model, and the second captures how much the head affects the model's output (good or bad). In Section 3.3 we only reported the recovered effect from KL divergence. We can also compute analogous quantities to Eqn. (3) for loss, in two different ways.

Following the ablation metric definition in Section 3.3.1, suppose at one token completion GPT-2 Small usually has loss $L$, though if we ablate of L10H7's direct effect has loss $L_{\text{Abl}}$. Then we could either measure $L_{\text{Abl}} - L$ and try and minimise the average of these values over the dataset, or we could instead

---

[7]As a concrete example, Elhage et al. (2021) show that a zero-layer transformer with tied embeddings cannot perfectly model bigrams in natural language.

[8]Thanks to an anonymous colleague for originally finding this result.

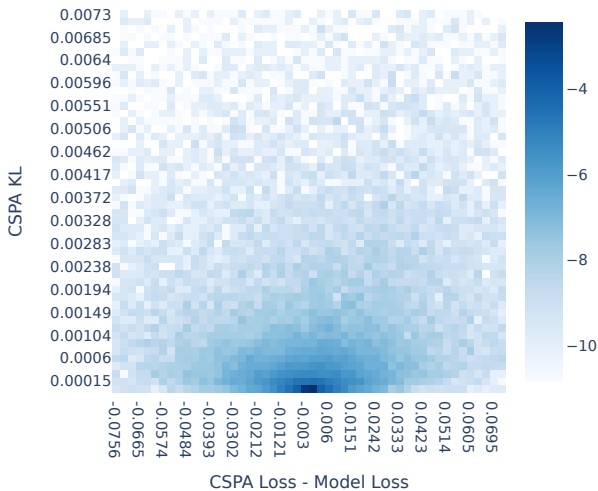

Figure 14: Log densities of dataset examples with loss change due to CSPA ($x$ axis) and KL divergence due to CSPA ($y$ axis). The $x$ axis range is between $-1$ and $+1$ standard deviation of loss changes due to CSPA, and the $y$ axis range is between $0$ and $+1$ standard deviation of CSPA KL.

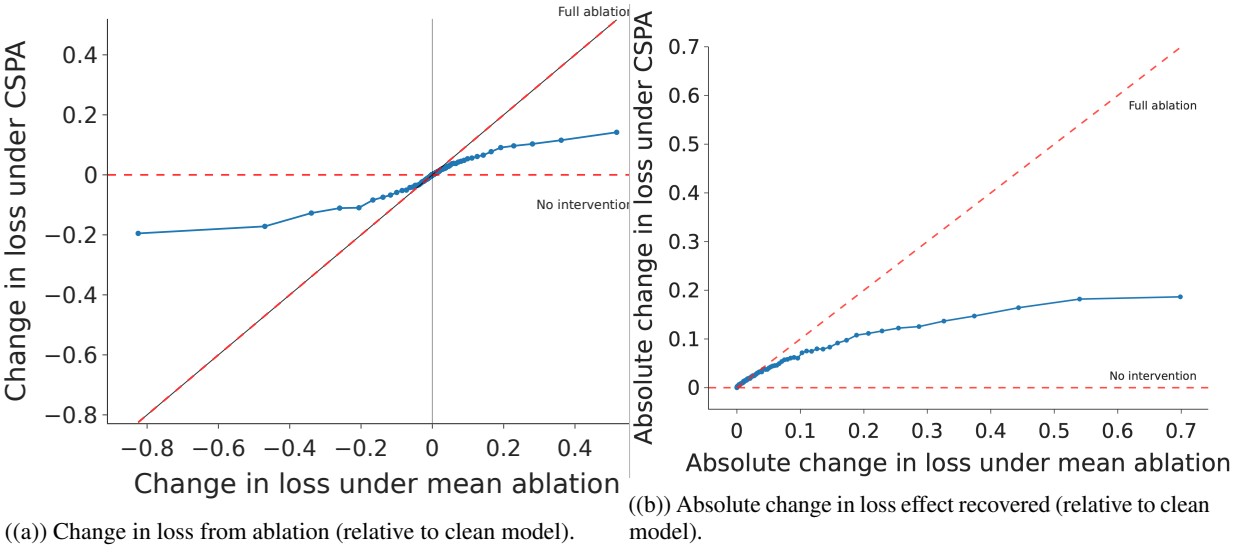

((a)) Change in loss from ablation (relative to clean model).

((b)) Absolute change in loss effect recovered (relative to clean model).

Figure 15: Studying CSPA under metrics other than KL Divergence.

minimize $|L_{\text{Abl}} - L|$. Either way, we can compare CSPA (Abl = CSPA) to the baseline of mean ablation (Abl = MA), by a similar ratio calculation as Eqn. (3). We get 82% effect recovered for the net loss effect and 45% effect recovered for the absolute change in loss. Despite these differing point values, the same visualisation method as Section 3.3.2) can be used to see where Copy Suppression is not explaining L10H7 behavior well (see Figure 15). We find that the absolute change in loss captures the majority of the model's (73.3%) in the most extreme change in loss percentile (Figure 15(b), far right), which shows that the heavy tail of cases where L10H7 is not very useful for the model is likely the reason for the poor performance by the absolute change in loss metric.

Also, surprisingly Figure 15(a)'s symmetry about $x = 0$ shows that there are almost as many completions on which L10H7 is harmful as there are useful cases. We observed that this pattern holds on a random sample of OpenWebText for almost all Layer 9-11 heads, as most of these heads have harmful direct effect on more than 25% of completions, and a couple of heads (L8H10 and L9H5) are harmful on the majority of token completions (though their average direct effect is beneficial).

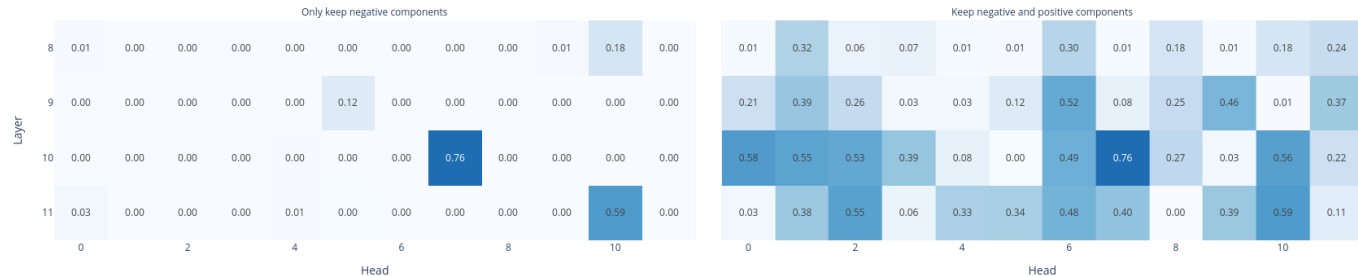

Figure 16: Calculating CSPA (with KL divergence) for all Layer 9-11 heads in GPT-2 Small.

### I.3 Does Eqn. (3) accurately measure the effect explained?

If Eqn. (3) is a good measure of the copy suppression mechanism, it should be smaller for heads in GPT-2 Small that aren't negative heads. We computed the CSPA value for all heads in Layers 9-11 in Figure 16.[9] We also ran two forms of this experiment: one where we projected OV-circuit outputs onto the unembeddings (right), and one where we only kept the negative components of OV-circuit outputs (left).

While we find that CSPA recovers more KL divergence L10H7 than all other heads, we also find that the QK and OV ablations (Section 3.3.1) lead to large ($> 50\%$) KL divergence recovered for many other heads, too. In ongoing experiments, we're looking into projection ablations on the QK circuit that will likely not recover as much KL divergence for other heads.

## J Semantic Similarity

42.00% of (source, destination) pairs had the source token in the top 10 most suppressed tokens, but not the most suppressed. When we inspect these cases, we find a common theme: the most suppressed token is often semantically related to the source token. For our purposes, we define **semantically related** as an equivalence relation on tokens, where if tokens $S$ and T are related via any of the following:

- Capitalization (e.g. " pier" and " Pier" are related),

- Prepended spaces (e.g. " token" and "token" are related),

- Pluralization (e.g. " device" and " devices" are related),

- Sharing the same morphological root (e.g. "drive", "driver", "driving" are all related)

- Tokenization (e.g. " Berkeley" and "keley" are related, because the non-space version "Berkeley" is tokenized into ["Ber", "keley"]).

We codify these rules, and find that in 90% of the aforementioned cases, the most suppressed token is semantically related to the source token. Although part of this is explained by the high cosine similarity between semantically related tokens, this isn't the whole story (on this set of examples, the average cosine similarity between the source token and the semantically related most suppressed token was 0.520). We speculate that the copy suppression algorithm is better thought of as **semantic copy suppression**, i.e. all tokens semantically related to the source token are suppressed, rather than **pure copy suppression** (where only the source token is suppressed). The figure below presents some OpenWebText examples of copy suppression occurring for semantically related tokens.

---

[9]All attention heads in Layers 0-8 have small direct effects: the average increase in loss under mean ablation of these direct effects is less than 0.05 for all these heads, besides 8.10. However heads in later layers have much larger direct effects, e.g 10/12 attention heads in Layer 10 (including L10H7) have direct effect more than 0.05.

Table 3: Dataset examples of copy suppression, with semantic similarity.

| Prompt | Source token | Incorrect completion | Correct completion | Form of semantic similarity |
|---|---|---|---|---|
| ...America's private **prisons** ... the biggest private **prison** - ... | " prisons" | " prison" | "-" | Pluralization |
| ...Steam**VR** (formerly known as Open**VR**), Valve's alternate **VR reality** ... | "VR" | " VR" | " reality" | Prepended space |
| ...Ber**keley** to offer course ... university of **Berkeley California** ... | "keley" | " Berkeley" | " California" | Tokenization |
| ...**Wrap** up the salmon fillets in the foil, carefully **wrapping sealing** ... | " Wrap" | " wrapping" | " sealing" | Verb conjugation & capitalization |

## K  Breaking Down the Attention Score Bilinear Form

In Section 4, we observed that Negative Heads attend to IO rather than S1 due to the outputs of the Name Mover heads. We can use QK circuit analysis (Section 3.2) in order to understand what parts of L10H7's query and key inputs cause attention to IO rather than S1.

As a gentle introduction to our methodology in this section, if an attention score was computed from an incoming residual stream vector $q$ at queryside and $k$ at queryside, then mirroring Eqn. (2) we could decompose the attention score

$$s = q^\top W_{QK}^{\text{L10H7}} k \tag{5}$$

into the query component from each residual stream component[10] (e.g MLP9, the attention heads in layer 9, ...) so $s = q_{\text{MLP9}}^\top W_{QK}^{\text{L10H7}} k + q_{\text{L9H0}}^\top W_{QK}^{\text{L10H7}} k + \cdots$. We could then further decompose the keyside input in each of these terms.

However, in this appendix we're actually interested in the difference between how the model attends to IO compared to S, so we decompose the attention score difference

$$\Delta s := q^\top W_{QK}^{\text{L10H7}} k^{\text{IO}} - q^\top W_{QK}^{\text{L10H7}} k^{\text{S1}} = q^\top W_{QK}^{\text{L10H7}} (k^{\text{IO}} - k^{\text{S1}}). \tag{6}$$

Since $\Delta s$ is in identical form to Equation (5) when we take $k = k^{\text{IO}} - k^{\text{S1}}$, we can decompose both the query inputs and key inputs of $\Delta s$. We also take $q$ from the END position in the IOI task. Under this decomposition, we find that the most contributions are from L9H6 and L9H9 queryside and MLP0 keyside (Figure 17(a)), which agrees with our analysis throughout the paper.

Further, we can test the hypotheses in Section 3.1 and Section 3.2 that copy suppression is modulated by an unembedding vector in the residual stream, by further breaking up each of the attention scores in Figure 17(a) into 4 further components, for the queryside components parallel and perpendicular to the unembedding direction, as well as the keyside components parallel and perpendicular to the MLP0 direction (Figure 17(b)). Unfortunately the direction perpendicular to IO is slightly more important than the parallel direction, for both name movers. This supports the argument in Section 4 that self-repair is more general than the simplest possible form of copy suppression described in Section 3.2.

## L  L10H7's QK-Circuit

### L.1  Details on the QK-Circuit experiments (Figure 3).

We normalize the query and key inputs to norm $\sqrt{d_{\text{model}}}$ to simulate the effect of Layer Norm. Also, MLP0 in Figure 3 refers to taking the embeddings for all tokens and feeding this through MLP0 (so is identical to effective embedding besides not having $W_E$ added).

---

[10]As in Eqn. (2), we found that the query and key biases did not have a large effect on the attention score difference computed here.

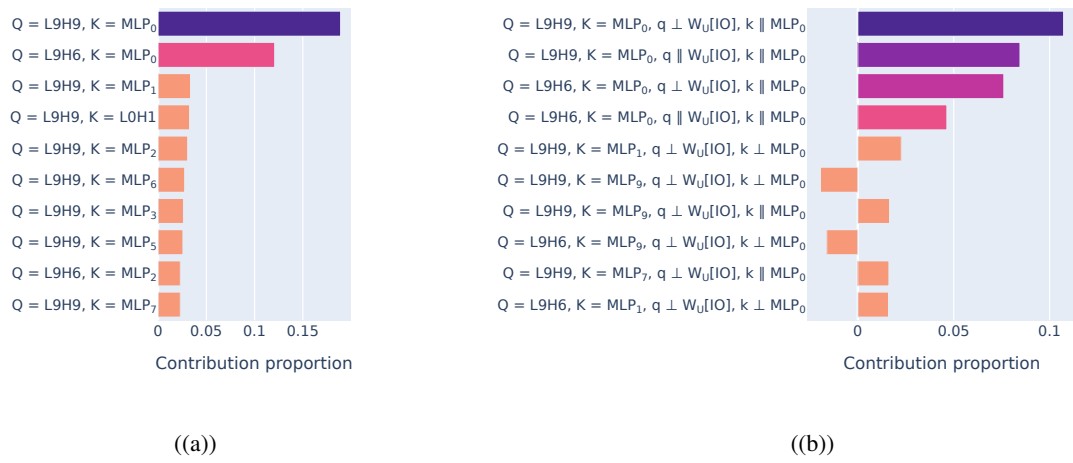

((a))                                  ((b))

Figure 17: Decomposing the bilinear attention score. 17(a): decomposing by all model components. 17(b): decomposing by all model components, and further by terms in the MLP0 direction (keyside) and terms in the IO unembedding direction (queryside). Terms involving name movers and MLP0 are highlighted.

Actually, key and query biases don't affect results much so we remove them for simplicity of Eqn. (2). Results when we uses these biases can be found in Figure 18(a). Additionally, the median ranks for other attention heads do not show the same patterns as Figure 3: for example, Duplicate Token Heads (Wang et al., 2023) have a 'matching' QK circuit that has much higher median ranks when the queryside lookup table is an embedding matrix (Figure 18(b)). Additionally, most other attention heads are different to copy suppression heads and duplicate token heads, as e.g for Name Mover Heads across all key and queryside lookup tables the best median rank is 561.

## L.2   Making a more faithful keyside approximation

Is our minimal mechanism for Negative Heads faithful to the computation that occurs on forward passes on dataset examples? To test this, we firstly select some important key tokens which we will measure faithfulness on. We look at the top 5% of token completions where L10H7 was most useful (as in Section 2) and select the top two non-BOS tokens in context that have maximal attention paid to them. We then project L10H7's key input onto a component parallel to the effective embedding for the key tokens, and calculate the change in attention paid to the selected key tokens. The resulting distribution of changes in attention can be found in Figure 19.

We find that the median attention change is $-0.09$, with lower quartile $-0.19$. Since the average attention amongst these samples is $0.21$, this suggests that the effective embedding does not faithfully capture the model's attention.

To use a more faithful embedding of keyside tokens, we run a forward pass where we set all attention weights to tokens other than BOS and the current token to 0. We then measure the state of the residual stream before input to Head L10H7, which we call the **context-free residual state**. Repeating the experiment used to generate Figure 19 but using the context-free residual state rather than the effective embedding, we find a more faithful approximation of L10H7's keyside input as Figure 20 shows that the median change in L10H7's attention weights is $-0.06$ which is closer to 0.

## L.3   Making a more faithful queryside approximation

We perform a similar intervention to the components on the input to the model's query circuit. We study the top 5% of token completions where L10H7 has most important effect. For the two key tokens with highest attention weight in each of these prompts, we project the query vector onto the unembedding vector for that key token. We then recompute attention probabilities and calculate how much this differs from the unmodified model. We find that again our approximation still causes a lot of attention decrease

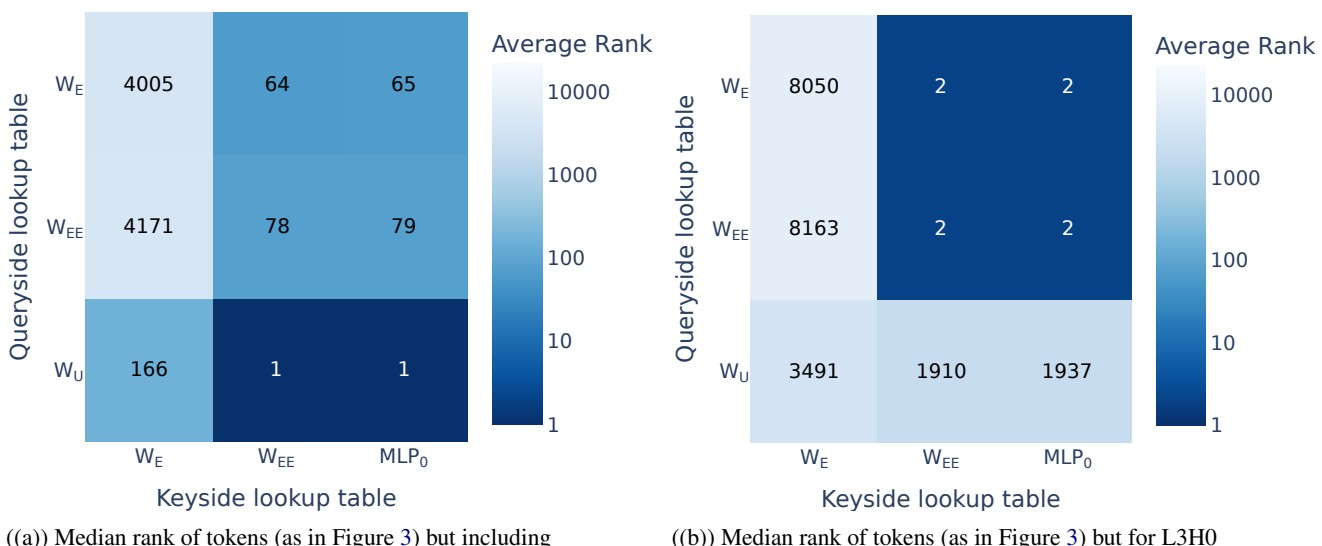

((a)) Median rank of tokens (as in Figure 3) but including biases before multiplying query and key vectors.

((b)) Median rank of tokens (as in Figure 3) but for L3H0 (a Duplicate Token Head).

Figure 18: Median rank of tokens (as in Figure 3) while adding biases (Figure 18(a)) and on a different head (Figure 18(b))

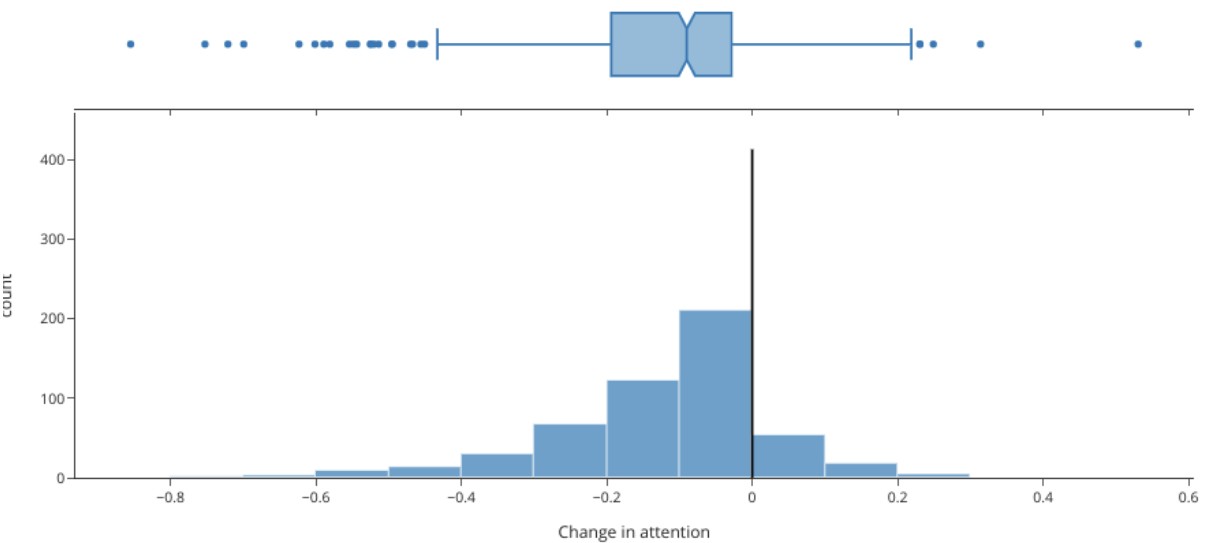

Figure 19: Change in attention on tokens when projecting key vectors onto the effective embedding for tokens.

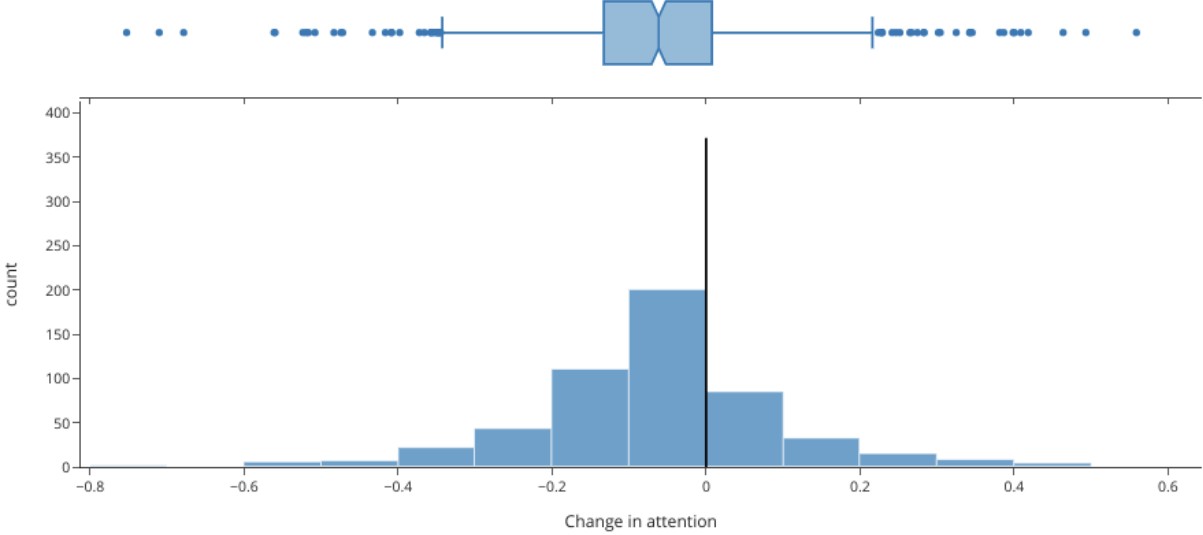

Figure 20: Change in attention on tokens when projecting key vectors onto the context free residual state.

in many cases (Figure 21).

There is a component of the queryside input perpendicular to the unembedding direction that is important for L10H7's attention. This component seems more important for L10H7s attention when the unembedding direction is more important, by performing an identical experiment to the experiment that produced Figure 21 except projecting onto the perpendicular direction, and then measuring the correlation between the attention change for both of these interventions on each prompt, shown in Figure 22. The correlation shows that it's unlikely that there's a fundamentally different reason why L10H7 attends to tokens other than copy suppression, as if this was the case it would be likely that some points would be in the low very negative $x$, close-to-0 $y$ region. This does not happen often.

We're not sure what this perpendicular component represents. Appendix R dives deeper into this perpendicular component in the IOI case study, and Appendix K further shows that the model parts that output large unembedding vectors (the Name Mover heads) are also the parts that output the important perpendicular component.

## M  CSPA with query projections

In this appendix, we design a similar ablation to CSPA, except we compute L10H7's attention pattern by only using information about the unembeddings in the residual stream, and the exact key tokens present in context, and we also do not perform any OV interventions. This means that together we only study how confident predictions in the residual stream are, as well as which types of tokens are more likely to be copy suppressed.

**A simple baseline.** The simplest query projection intervention is to recalculate the attention score on each key token $T$ by solely using the residual stream component in the direction $W_U[T]$. Sadly, this intervention results in only 25% of KL divergence recovered.

**Improving the baseline.** Observing the starkest failure cases of the simple baseline, we often see that this intervention neglects cases where a proper noun and similar words are copy suppressed: the model attended most to a capitalized word in context 9x times as frequently as occurred in this ablation. To

remedy these problems, we performed two changes. 1) Following Appendix J, when we compute the attention score back to a token $T$, we don't just project onto the unembedding vector $W_U[T]$, but instead take all $T^*$ that are semantically similar to $T$, and project onto the subspace spanned by all those vectors.

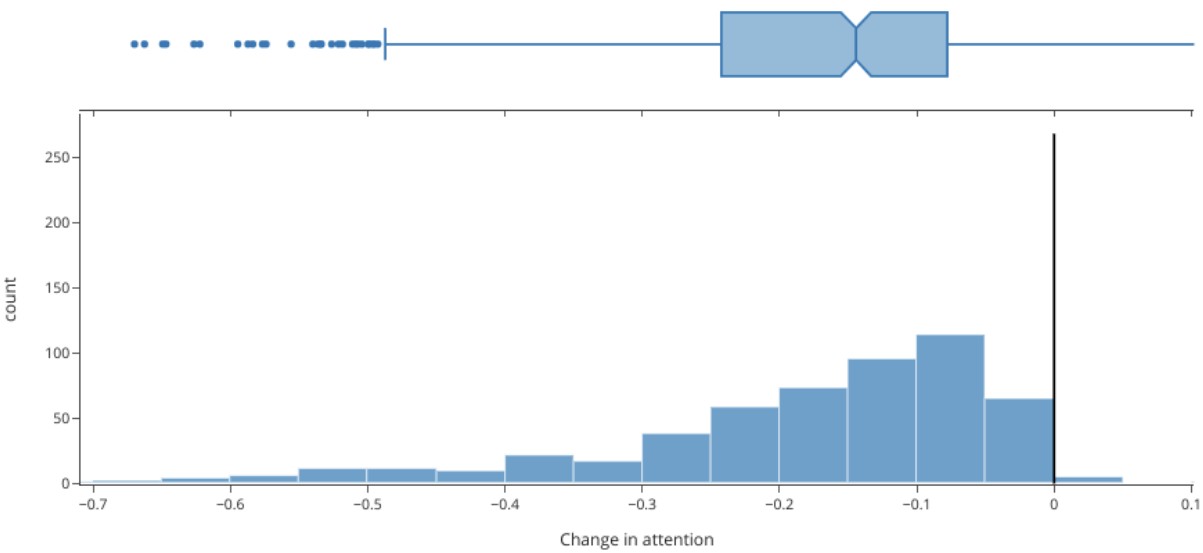

Figure 21: Change in attention on tokens when projecting query vectors onto the unembedding vectors for particular tokens.

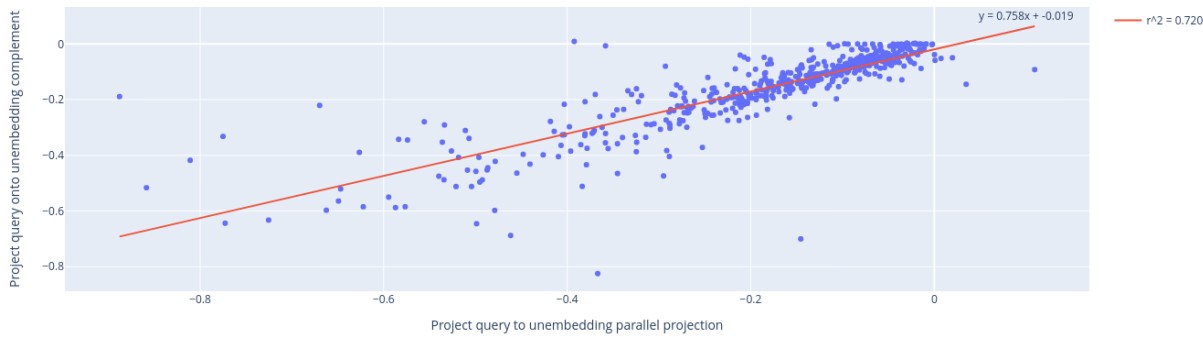

Figure 22: Correlation between change in attention on tokens when projecting onto the component parallel to the unembedding and ($x$-axis) and also projecting onto the component perpendicular to the unembedding ($y$-axis).

2) we learnt a scaling and bias factor for every token in GPT-2 Small's vocabulary, such that we multiply the attention score back to a token $T$ by the scaling factor and then add the bias term. We never train on the test set we evaluate on, and for more details see our Github (which will be released upon successful publication). With this setup, we recover 61% of KL divergence.

**Limitations.** This setup may recover more KL divergence than the 25% of the initial baseline, but clearly shows that L10H7 has other important functions. However, observing the cases where this intervention has at least 0.1 KL divergence to the original model (57/6000 cases), we find that in 39/57 of the cases the model had greatest attention to a capitalized word, which is far above the base rate in natural language. This suggests that the failure cases are due to our projection not detecting cases where the model should copy suppress a token, rather than L10H7 performing an entirely different task to copy suppression.

## N  Weights-based evidence for self-repair in IOI

In this section, we provide evidence for how the attention heads in GPT-2 Small compose to perform self-repair. As shown in Elhage et al. (2021), attention heads across in different layers can compose via the residual stream.

Copy Suppression qualitatively explains the mechanism behind the self-repair performed in the Negative Heads: ablating the upstream Name Mover Heads reduces copying of the indirect object (IO) token, causing less attention to that token (Appendix O). In this section, we show that the opposite effect arises in backup heads: ablation indirectly cause more attention to the IO token, as the Name Mover Heads outputs prevent backup heads from attending to the IO token.

To reach this conclusion, we conduct a weights-based analysis of self-repair in GPT-2 Small. Specifically, we can capture the reactivity of downstream heads to Name Mover Heads by looking at how much the OV matrix $W_{OV}$ of the Name Mover Heads causes Q-composition (Elhage et al., 2021) with the QK matrix $W_{QK}$ of a downstream QK-head. To this end, we define

$$M := \text{MLP}_0(W_E)^\top W_{OV}^T W_{QK} \text{MLP}_0(W_E) \in \mathbb{R}^{n_{\text{vocab}} \times n_{\text{vocab}}}. \tag{7}$$

$M$ is an extension to the setup in Section 3.2.[1112] We studied this composition over the $n_{\text{names}} = 141$ name tokens in GPT-2 Small's vocabulary by studying the $\mathbb{R}^{n_{\text{names}} \times n_{\text{names}}}$ submatrix of $M$ corresponding to these names. For every (Name Mover Head, QK-head) pair, we take the submatrix and measure the median of the list of ranks of each diagonal element in its column. This measures whether QK-heads attend to names that have been copied by Name Movers (median close to 1), or avoid attending to these names (median close to $n_{\text{names}} = 141$). Figure 23 shows the results.

These ranks reflect qualitatively different mechanisms in which self-repair can occur (Table 2). In the main text Figure 26, we colour edges with a similar blue-red scale as Figure 24.

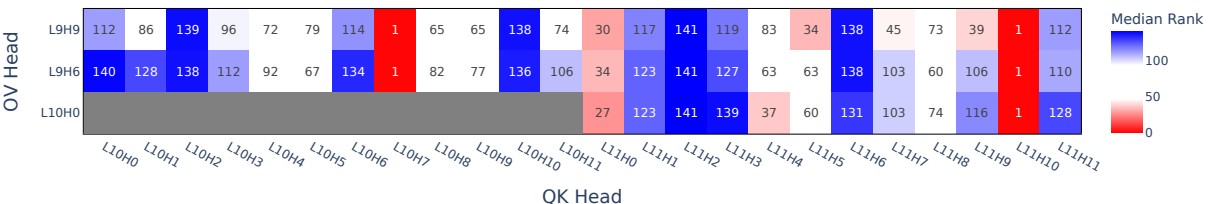

Figure 23: A graph of the Median Token Ranks between the Name Mover Heads (on the OV side) and Layer 10 and 11 Heads (on the QK side), to measure $Q$-composition in the $QK$ circuit. There are $n_{\text{names}} = 141$ names.

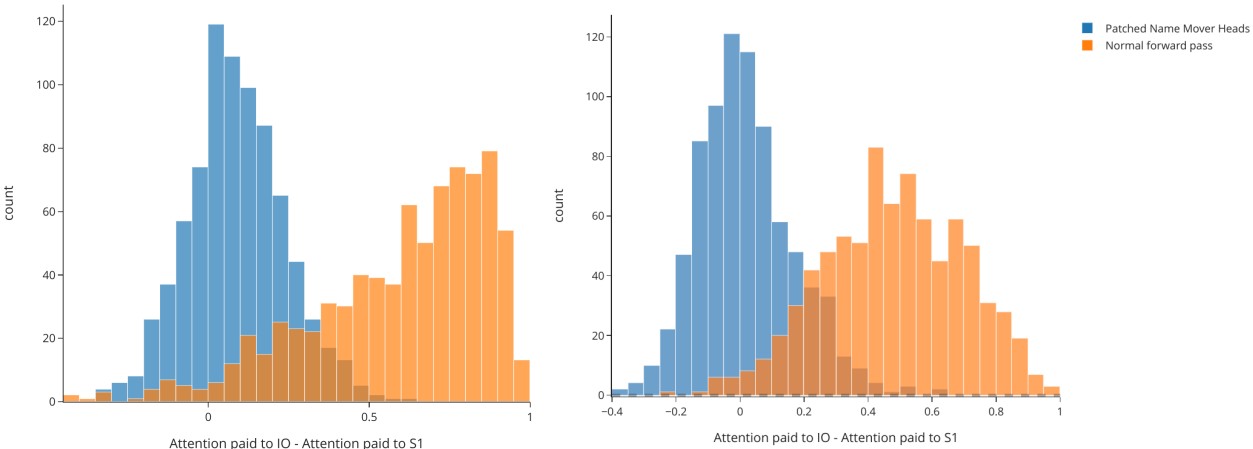

Figure 24: Measuring the difference in attention paid to different names when editing the input Negative Heads receive from Name Mover Heads.

## O    Negative heads' self-repair in IOI

We edited the input that the Negative Heads receive from the Name Mover heads by replacing it with an activation from the ABC distribution. We then measured the difference between the attention that the negative head paid to the IO token compared to the S token. We found that the Negative Heads now attended equally to the IO and the S1 token, as the average IO attention minus S1 attention was just 0.08 for Head L10H7 and 0.0006 for Head L11H10 (Figure 24).

Since Negative Heads are just copying heads (Section 3.1), this fully explains copy suppression.

## P    Universality of IOI Self-Repair

Since Negative Heads exist across distributions and models, we also expect that IOI self-repair potentially exists universally as well. Initial investigations across other models about self-repair in the IOI task highlight similarities to the phenomena we observe here but with some subtleties in the specifics. For instance, one head in Stanford GPT-2 Small E wrote 'less against' the correct token upon the ablation of earlier Name Mover Heads; however, it is distinct from the copy suppression heads in GPT-2 Small in that it attended to both the IO and S2 tokens equally on a clean run.

## Q    Amplifying Query Signals into Self-Repair Heads

As a part of our exploration into how self-repair heads respond to signals in the residual stream, we noticed that the output of the name mover heads was extremely important for the queries of the self-repair heads. We wanted to decompose the signal down into subcomponents to determine which parts were meaningful - in particular, we were curious if the IO unembedding direction of the name mover head's output was important.

To do this, we intervened on the query-side component of a self-repair head by:

1. Making a copy of the residual stream before the self-repair head, and adding a scaled vector $s\vec{v}$ (where $\vec{v}$ is a vector and $s$ is some scaling) to this copy (before the LayerNorm)

2. Replacing the query component of the head with the query that results from the head reading in this copied residual stream into the query

3. Varying the scaling $s$ while repeatedly observing the new attention patterns of the self-repair of the head

---

[11]This is similar to how Elhage et al. (2021) test the 'same matching' induction head QK circuit with a K-composition path through a Previous Token Head

[12]As in Section 3.2 we ignore query and key biases as they have little effect.

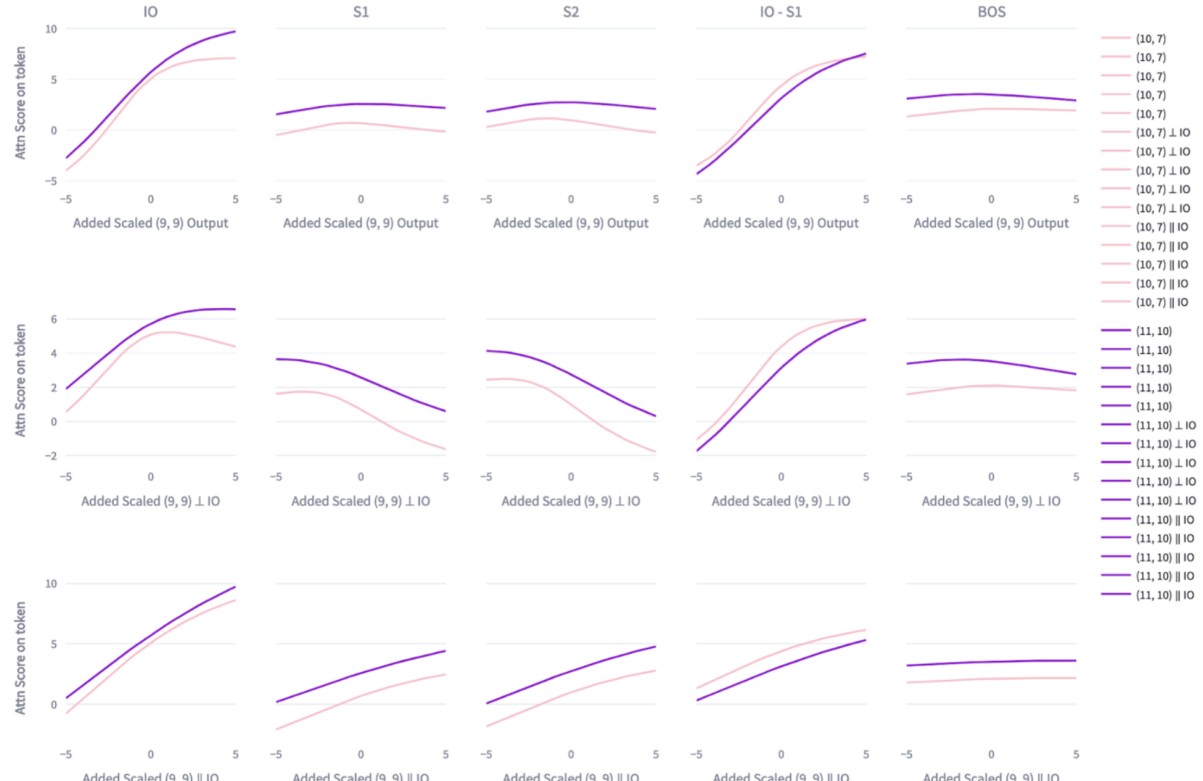

Figure 25: Observing the change in attention scores of Negative Heads upon scaling the presence of the output of L9H9, both parallel and perpendicular to the $W_U[IO]$ direction.

Figure 25 shows a specific instance in which the vector is the output of head L9H9. We add scaled versions of the output into the residual streams of the Negative Heads which produce their queries (before LayerNorm). Additionally, we do an analogous operation with the projection of L9H9 onto the IO unembeddings, as well as the projection of L9H9 away from the IO unembeddings.

We observe that the Negative Heads have a positive slope across all of the IO subgraphs. In particular, this still holds while using just the projection of L9H9 onto the IO unembedding direction: this implies that the greater the presence of the IO unembedding in the query of the negative name mover head, the greater the neagtive head attends to the IO token. The result still holds whether or not we add the vector before or after LayerNorm, or whether or not we freeze LayerNorm.

Unfortunately, this same trend does not hold for backup heads. In particular, it seems that while we expect a predictable 'negative' slope of all the subgraphs (as the L9H9 output causes the backup heads to attend less to the IO token), this trend does *not* hold for the projection of L9H9 onto the IO unembedding. This provides additional evidence for the claim that the unembeding component is not the full story of all of self-repair.

## R   Complicating the Story: Component Intervention Experiments

Copy suppression explains self-repair in negative heads via the importance of the unembedding direction (Section 3.2). Ideally, the unembedding direction would also help understand backup heads. However, we present two pieces of evidence to highlight how the unembedding only explains part of the self-repair in GPT-2 Small, including showing that our understanding of Negative Heads on the IOI task also requires understanding more than simply the unembedding directions.

First, we intervened on the output of the Name Movers and L10H7,[13] and edited the resulting changes into the queries of downstream heads. The intervention, shown in Figure 26, was either a projection *onto*

---

[13]We also ablate the output of L10H7 due to self-repair that occurs between L11H10 and L10H7, as explained in Appendix B.

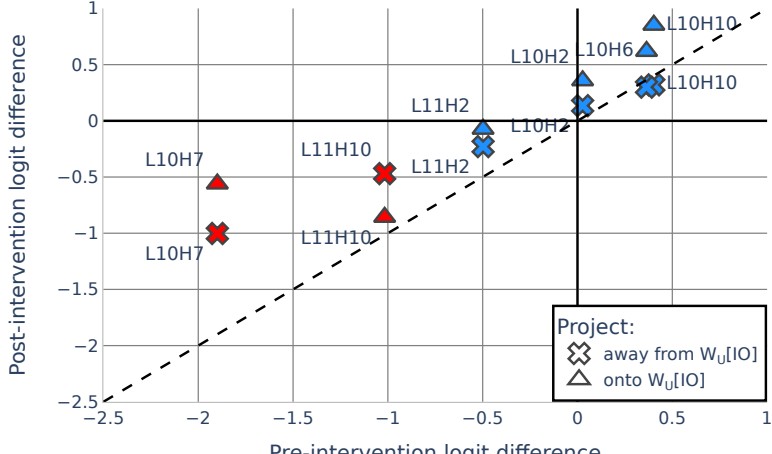

Figure 26: Intervening in the IO unembedding input into self-repairing heads, and measuring the logit difference before and after these intervetions. The unembedding direction doesn't completely describe the backup effect.

or *away from* the IO unembedding $W_U[\text{IO}]$[14]. We also froze the Layer Norm scaling factor equal to the value on the original forward pass. To interpret Figure 26, note that for most backup heads, projecting away from $W_U[\text{IO}]$ does not change the heads' logit differences much, suggesting that the unembedding direction isn't very causally important for self-repair in backup heads. As such, there must be important information in the $W_U[\text{IO}]$-perpendicular direction that controls self-repair.

To complement this analysis, we also broke the attention score (a quadratic function of query and key inputs) down into terms and again found the importance of the perpendicular direction (Appendix K). Beyond this, intervening in the queries of self-repair heads reflects that the perpendicular direction is particularly important in the Backup Heads (Appendix Q). Ultimately, we conclude that while Name Mover Heads modulate Negative Heads' copy suppression, this is only partly through the unembedding direction. Further, backup heads do not seem to depend on the unembedding direction.

---

[14]By 'away from', we mean removing the unembedding direction from the head output, so the resultant vector is orthogonal to the unembedding direction.

