# OpenReview forum: "Copy Suppression: Comprehensively Understanding a Motif in Language Model Attention Heads"
_EMNLP/2024/Workshop/BlackBoxNLP — BlackboxNLP 2024_

### Official Review · Reviewer_5Z8R · 2024-09-05

**Overall Assessment:** 4
**Confidence:** 3

**Best Paper:**

2

**Best Paper Justification:**

Discovery and thorough analysis of a new and interesting behavior of language model attention heads.

**Comments Questions Suggestions And Typos:**

-

**Paper Summary:**

The paper thoroughly examines "copy suppression", a behavior of attention heads in language models that suppresses the prediction of previously generated tokens. The authors find that Layer 10 Head 7 (L10H7) in GPT-2 small implements this algorithm, and that 76.9% of its behavior on the training distribution is copy suppression. Furthermore, the authors demonstrate that copy suppression plays a critical role in anti-induction and self-repair, and that copy suppression occurs in different model architectures too. The authors claim that they provide the most comprehensive description of a language model component to date.

**Summary Of Strengths:**

I think the paper has many strengths:
- Novelty: The paper analyses a previously unexplored behavior of attention heads (copy suppression);
- Thoroughness: The analyses are very detailed and convincing;
- Cross-model generalisation: The authors demonstrate the copy suppression occurs in multiple model architectures (GPT-2, Pythia, and SoLU), which is particularly interesting, because the mechanistic interpretability research that I've seen before usually only focuses on a single model;
- Demo: The authors provide a demo webpage with nice visualizations and examples to illustrate their findings.

**Summary Of Weaknesses:**

I would say that the main weakness of the paper is how the results are presented (although I really appreciate the demo). At times, the technical explanation can be difficult to follow, especially for readers not deeply familiar with mechanistic interpretability techniques. This could make the paper less accessible to a broader audience.

---

### Official Review · Reviewer_368w · 2024-09-06

**Overall Assessment:** 4
**Confidence:** 4

**Best Paper:**

2

**Best Paper Justification:**

Definitely a very thorough and impressive piece of mech interpretability, but it is dense and a bit of an information overload.

**Comments Questions Suggestions And Typos:**

* Line 86: " war" -> "war"
* Section 2.2 was very terse, and Figure 2 somewhat hard to interpret without more guidance.  I think I was able to figure out most of what's happening here, but it could and should definitely be expanded a bit.
* Line 227 ends in the middle of a sentence.
* Line 267: "(1))" -> "(1)"
* Section 3.1, OV circuit: does it matter that this circuit is being applied to the "effective embedding", and not the actual input to the attention head, which is in Layer 10?
* How was L10H7 discovered in the first place? I appreciate the detailed analysis of its behavior, but something more could be said about how this particular head was first discovered for analysis.
* Line 312: missing a period after the parenthetical comment.
* Figure 4: what exactly do the dashed and solid lines mean? Same for shading; This figure could be labeled / annotated a bit more thoroughly to guide the reader.
* Section 3.3.2: it seems surprising to me that both the OV and QK ablations explain more of L10H7 alone than the copy suppression ablation does.  How do you interpret this?  It seems like we should say this head does the "QK thing" more than copy suppression, since it explains the most.
* Line 422 links to a glossary item, which just says "look at the sentence you just clicked from".  This was odd.  I'd keep the glossary entry, but remove this particular link to it.
* Line 444: missing a closing parenthesis.

**Paper Summary:**

This paper introduces a mechanism---copy suppression---that can be implemented by an attention head and argue that one such head in GPT2-Small (L10H7) does in fact implement this mechanism.  The mechanism effectively down-weights output probabilities ("suppresses") assigned to tokens that have already appeared in the context (which would be "copies").  This represents a very detailed analysis of an attention head in a model trained on naturalistic data as opposed to a small toy model.  The authors also go on to show that some previously identified attention head behaviors in the literature can be helpfully analyzed through the lens of copy suppression.  The paper makes a very valuable contribution to the mechanistic interpretability literature, and should be of interest to a large portion of the Blackbox audience, so I'd like to see it appear.  There are a few questions I have about how to understand the results and the take-home message, which are mentioned below.

**Summary Of Strengths:**

* Identifies a new behavioral mechanism (copy suppression) that is used to explain an attention head in GPT2-small, a model trained on natural data.  This is a detailed analysis of a head in a more natural setting than a lot of mechanistic interpretability literature.
* Shows how some previously identified mechanisms can be understood as instances of copy suppression.
* Overall well-written.  I especially appreciated the use of a linked glossary for terms that are somewhat common in the mechanistic interpretability literature but not in the wider analysis and interpretability space.

**Summary Of Weaknesses:**

* Unclear whether copy suppression is the "best" explanation of L10H7, given the results in 3.3.2 which show that other ablations explain more of the head's behavior.
* Something of a "too much information" phenomenon.  The authors have done so many experiments and so many analyses, that it becomes hard to follow in places and some discussions are not had at a sufficient depth (see comments below).  This could certainly be more than one paper.

---

### Official Review · Reviewer_qGQk · 2024-09-12

**Overall Assessment:** 4
**Confidence:** 5

**Best Paper:**

1

**Best Paper Justification:**

N/A

**Comments Questions Suggestions And Typos:**

L345: What does it mean to keep only the negative components? I found this a little vague.

For the QK ablation, you are taking the logit lens of the destination token at the hidden state L (L=10 when analyzing 10.7)?

I think the graphics for Figure 4 do not help illustrate the textual description. I think it would be helpful to see things actually being ablated

Have the authors considered the connection to inhibition heads from Wang et al., 2023? They are a fundamentally different mechanism but accomplish the same/similar things. I'd be curious to know if negative movers and inhibition heads are always active at the same time as negative movers


* Does a word *have* to be in the context for it to be suppressed?

**Paper Summary:**

This paper investigates a common pattern in the role of an attention head in GPT-2: copy suppression, which is used to demote the probability of a certain word which has already been seen in context. These heads will typically suppress the wrong answer to prompts (such as in the IOI task), or what the authors call "naive" copying, where repetitive continuations are downweighted: e.g., "Beijing is located in" --> Beijing.
Through extensive experiments, the authors boil down the role of this head to this specific operation, and show that ablating it has a significant direct effect on the logits for relevant inputs. The copy suppression motif is strengthened through weight analysis, which shows that this head is specifically 'designed' for this role. The evidence for this is through vocab projections, where it's shown that Queries will disproportionately attend to Keys corresponding to the same exact token (A attends to A, B attends to B) in the negative mover head. The authors build on this nicely to explain more complicated phenomena like backup heads (initially described in Wang et al., 2023) and self-repair, a mechanism in which perturbations earlier in the model modulate the activity of the negative mover head in the predicted way.

**Summary Of Strengths:**

* This paper tells a clear story about the role of an attention head, related heads, and a mechanism that involves them (self-repair). It contributes to a deeper understanding of how basic processing is carried out in a language model

* I like the analysis comparing ablations to ablating the direct effect of 10.7. This is a nice experiment that tests whether the hypothesized function explains the actual behavior of the head. My understanding is that this discounts indirect effects, but given the evidence of what the head is doing up to this point in the paper (Figure 2), this simplification doesn't matter much

This paper will be of interest to those in the mech interp community and builds on analysis/understanding of GPT2-small, so I'd like to see it accepted

**Summary Of Weaknesses:**

* Some figures/descriptions are hard to parse. See below.

* I would like to see more of the results for copy suppression in other models. Figure 5 does not really convey all of the essential information about this. Is there a method that we can use to detect copy suppression in other models? The authors discuss the weights based approach (vocab projections), but it isn't really clear whether this extends to other model families. Positive or negative results would be interesting and informative here, but the paper doesn't provide them. I believe analyzing and deeply understanding a single model should be sufficient, but the mechanism is simple to check (and to an extent the authors already have), so it seems like a worthwhile thing to include

---

### Decision · Program_Chairs · 2024-09-18

**Decision:**

Accept

**Comment:**

The paper carefully analyzes attention heads for a specific algorithm of "copy suppression" in a natural and controlled setting. All reviewers unanimously agree that the paper provides a detailed analysis and makes a valuable contribution to the interpretability community.